# The encephalomyocarditis virus Leader promotes the release of virions inside extracellular vesicles via the induction of secretory autophagy

Susanne G. van der Grein[1,4], Kyra A. Y. Defourny [1,4], Huib H. Rabouw[2], Soenita S. Goerdayal[3], Martijn J. C. van Herwijnen[1], Richard W. Wubbolts[1], Maarten Altelaar [3], Frank J. M. van Kuppeveld [2] & Esther N. M. Nolte-'t Hoen [1✉]

Naked viruses can escape host cells before the induction of lysis via release in extracellular vesicles (EVs). These nanosized EVs cloak the secreted virus particles in a host-derived membrane, which alters virus-host interactions that affect infection efficiency and antiviral immunity. Currently, little is known about the viral and host factors regulating this form of virus release. Here, we assessed the role of the encephalomyocarditis virus (EMCV) Leader protein, a 'viral security protein' that subverts the host antiviral response. EV release upon infection with wildtype virus or a Leader-deficient mutant was characterized at the single particle level using high-resolution flow cytometry. Inactivation of the Leader abolished EV induction during infection and strongly reduced EV-enclosed virus release. We demonstrate that the Leader promotes the release of virions within EVs by stimulating a secretory arm of autophagy. This newly discovered role of the EMCV Leader adds to the variety of mechanisms via which this protein affects virus-host interactions. Moreover, these data provide first evidence for a crucial role of a non-structural viral protein in the non-lytic release of picornaviruses via packaging in EVs.

[1] Division of Cell Biology, Metabolism & Cancer, Department of Biomolecular Health Sciences, Faculty of Veterinary Medicine, Utrecht University, Yalelaan 2, 3584 CM Utrecht, The Netherlands. [2] Virology Section, Division Infectious Diseases & Immunology, Department of Biomolecular Health Sciences, Faculty of Veterinary Medicine, Utrecht University, Yalelaan 1, 3584 CL Utrecht, The Netherlands. [3] Biomolecular Mass Spectrometry and Proteomics, Bijvoet Center for Biomolecular Research and Utrecht Institute for Pharmaceutical Sciences, Utrecht University, Padualaan 8, 3584 CH Utrecht, The Netherlands. [4]These authors contributed equally: Susanne G. van der Grein, Kyra A. Y. Defourny. ✉email: e.n.m.nolte@uu.nl

Picornaviruses form a large family of small (~30 nm) single-stranded positive-sense RNA viruses that have long been believed to be exclusively released from cells as non-enveloped or "naked" progeny by inducing cell rupture. However, we and others showed that several picornavirus species, including hepatitis A virus (HAV), poliovirus, enterovirus 71 (EV71), coxsackievirus B3 (CVB3), and encephalomyocarditis virus (EMCV), can also be released from cells in a non-lytic fashion via packaging in extracellular vesicles (EVs)[1–5]. EVs are lipid bilayer-enclosed particles with a predominant size range of 50-300 nm that are released by all types of cells and deliver protein or RNA cargo to distant or neighboring cells as a means of intercellular communication (reviewed in refs. [6,7]). The existence of EV-enclosed virus particles in vivo has been confirmed for several types of naked viruses, including picornaviruses, hepatitis E virus, rotavirus, and norovirus[1,4,8–10], and raises important questions regarding their role in infection. It has been well established that virus-containing EVs can transfer the infection to new host cells. Importantly, EV-enclosed viruses may differ in biological properties compared to their naked counterparts. Incorporation in EVs allows viruses to escape immune surveillance as the EV membrane shields virions from recognition by neutralizing antibodies[1,11–13]. Likewise, the EV membrane may alter cellular uptake dynamics and affect the biodistribution of the virus, including passage over barriers such as the blood-brain barrier[14–19]. In addition, EV-enclosure may increase replicative fitness of the virus through simultaneous delivery of multiple genetic quasi-species[3,10,20]. As a result, EV-enclosure of viruses can increase infection efficiency. Conversely, the transfer of virus or host molecules from infected cells via EVs may also trigger antiviral immune responses in targeted cells, thereby restricting the spread of the infection[21–23].

While several lines of evidence indicate the functional importance of EV-enclosed virus release, there are many unknowns regarding the formation of virus-containing EVs. The most common EV release pathways are the outward pinching of vesicles from the plasma membrane and the release of intraluminal vesicles via fusion of multivesicular endosomes (MVE) with the plasma membrane (reviewed in ref. [6]). Recently, several picornavirus species were reported to escape cells enwrapped in EVs containing the autophagy-regulatory protein LC3[2,3,5,24,25]. This finding triggered the idea that also autophagy could be involved in the formation of EV-enclosed viruses. Autophagy is a cellular process in which damaged or obsolete organelles, aggregated proteins, or intracellular pathogens are engulfed and broken down to recycle nutrients and maintain homeostasis[26]. While autophagy is often employed by host cells to defend themselves against invading viruses, several picornaviruses actively induce autophagy to support virus replication and release[27,28]. Based on the reported presence of LC3 on virus-containing EVs, it has been proposed that instead of following conventional routing towards lysosomes, autophagosomes that have engulfed nascent virions may expel their contents to the external milieu during viral infection. The process in which autophagosomal cargo is secreted into the extracellular environment instead of being degraded is generally referred to as "secretory autophagy". This process is known to facilitate unconventional secretion of cytosolic proteins lacking a signal peptide for the classical secretory pathway[29]. Although a role for autophagy in non-lytic virus release has been suggested, direct evidence for the specific involvement of a secretory arm of the autophagy pathway in the biogenesis and release of virus-containing EVs is currently lacking. In addition, it is unknown which viral and/or host factors are involved in regulating this form of virus release.

Here, we investigate whether and how viral proteins actively promote the release of EV-enclosed viruses by studying the Leader protein of encephalomyocarditis virus (EMCV), a model picornavirus with a rapid lytic life cycle. The EMCV Leader functions in the subversion of antiviral defense as well as the manipulation of host transcription and translation, and is therefore referred to as the main "viral security protein" of EMCV[30]. Previously, we demonstrated that EMCV induces the release of multiple EV populations and that only specific EV subpopulations were highly potent in spreading the infection[5]. We here investigate whether the EMCV Leader protein plays a role in the release of these infectious EVs using high-resolution flow cytometric quantification and characterization of single EVs. We demonstrate that the Leader protein acts as a major determinant for the release of virus-containing EV subpopulations. By tracking the fate of the autophagic marker LC3, we show that the Leader promotes the accumulation of autophagic structures that do not progress along the degradation pathway and that the Leader is required for the induction of secretory autophagy during infection. Pharmacological treatments validate that via the activation of secretory autophagy the Leader protein promotes the packaging of virus particles within EVs. These data exemplify the importance of non-structural viral proteins in promoting the release of EV-enclosed virions via modification of host cell secretory pathways.

## Results

**Inactivation of the Leader protein reduces EV-enclosed virus release by EMCV-infected cells.** The EMCV Leader is of critical importance for in vivo infection, yet was previously shown to be dispensable for virus replication[31]. Here, we set out to study the role of the Leader protein in the regulation of EV-enclosed virus release. Non-lytic release of virus and EVs was assessed in cells infected with either wildtype (Wt) EMCV or with a previously established mutant (EMCV-L$^{Zn}$) expressing a functionally inactive Leader protein[31]. Similar infection efficiencies were observed for both viruses upon infection at a high multiplicity of infection (MOI 10) (Supplementary Fig. 1). In addition, both viruses displayed similar replication kinetics during the pre-lytic phase of a single round of infection (Fig. 1a). Whereas only minor differences in total virus production were observed, EMCV-L$^{Zn}$ infected cells released substantially lower amounts of virus into the extracellular space than EMCV-Wt infected cells upon reaching a plateau in virus production (7–8 h p.i.) (Fig. 1b). Prolonged culture past 8 h p.i. led to virus-induced cell lysis for EMCV-Wt infected cells, whereas EMCV-L$^{Zn}$ infected cells could be cultured without loss of cell viability up to 12 h p.i., after which the induction of apoptotic cell death was observed, as reported previously[32]. However, even at this later time point virus release by EMCV-L$^{Zn}$ infected cells did not reach the level of virus release observed for EMCV-Wt at 8 h p.i. (Fig. 1b and Supplementary Fig. 2a). Virus packaging in EVs is believed to be an important form of virus release in the early stages of infection, which precedes the massive lysis-mediated release of naked virions during advanced infection[33–35]. Therefore, we investigated whether the defect in extracellular virus release for EMCV-L$^{Zn}$ infected cells was indicative of a defect in EV-enclosed virus release. We previously reported that EMCV-Wt infected cells release viruses within different EV subsets that can be isolated from cell culture supernatant by centrifugation at either 10,000×g (10 K) or 100,000×g (100 K) centrifugal force[5]. Hence, we compared the amount of virus incorporated in 10 K and 100 K EVs released during EMCV-Wt and EMCV-L$^{Zn}$ infection. EV-enclosed virus particles were separated from potentially co-isolating naked viruses based on buoyant density by means of isopycnic density gradient centrifugation[5]. Significantly more infectivity was detected in the 100 and 10 K EV isolates from EMCV-Wt compared to EMCV-L$^{Zn}$ infected samples (Fig. 1c), a difference that could not be explained

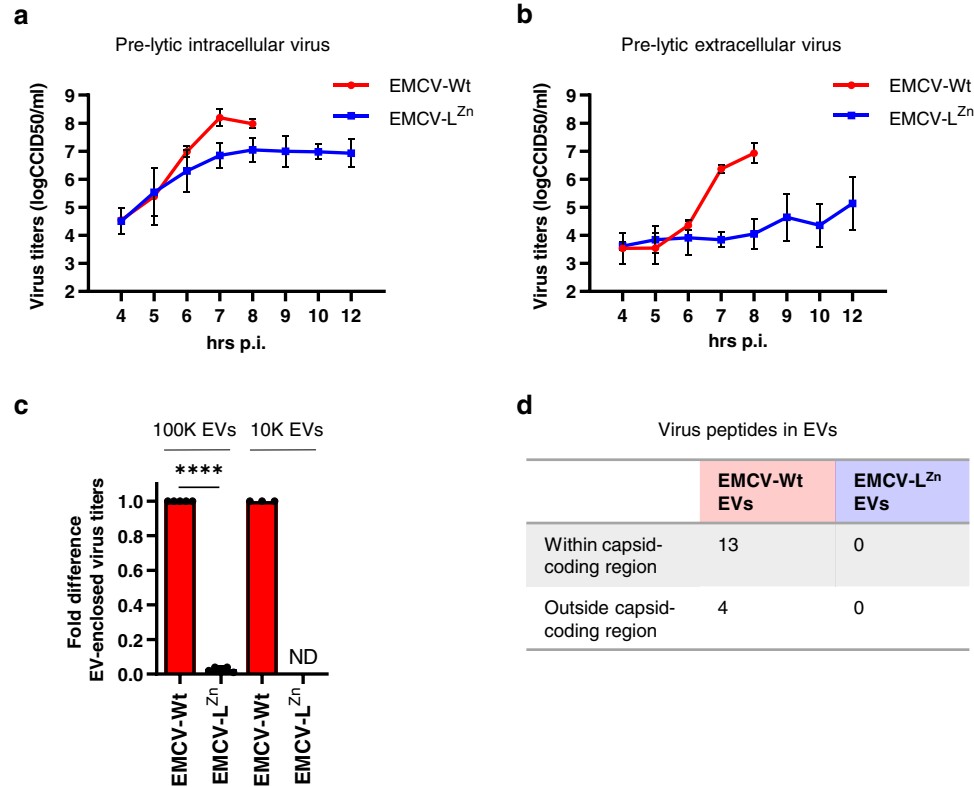

**Fig. 1 Inactivation of the Leader protein reduces EV-enclosed virus release by EMCV-infected cells. a, b** EMCV-Wt and EMCV-L$^{Zn}$-infected HeLa cells and corresponding cell culture supernatants were harvested at the indicated timepoints after infection. Line graphs display the increase in intracellular virus titers (**a**) and virus titers in the cell culture supernatant (**b**) over time, as determined by end-point dilution. Depicted are mean values ± SD from $n = 3$ independent experiments. **c** 10 K and 100 K EVs were separated from naked virions using density gradient centrifugation and EV-enclosed virus titers were determined by end-point dilution assay. Bars display the mean fold infectivity of EMCV-L$^{Zn}$ EVs relative to EMCV-Wt EVs ± SD from $n = 5$ independent experiments. ****$p < 0.0001$, as assessed by a two-tailed one-sample $t$-test. ND not detected. **d** 100 K EVs released by cells infected with either EMCV-Wt or EMCV-L$^{Zn}$ were isolated and EV-protein composition was analyzed by LC-MS/MS. The number of unique EMCV peptides identified within and outside of the capsid-coding region are indicated. Source data are provided as a Source Data file.

by differences in intracellular virus production (Supplementary Fig. 2b). Moreover, no infectivity could be retrieved in 10 K EV isolates of EMCV-L$^{Zn}$ cell supernatant (Fig. 1c). To validate that the decrease in infectivity in 100 K EVs was due to a lack of virus incorporation into EVs, we performed LC-MS/MS analysis of the 100 K EV samples and mapped the resulting peptide hits against the EMCV polyprotein. Seventeen different unique viral peptides were identified across the three replicate preparations of EMCV-Wt EVs (Fig. 1d). Most (13/17) unique peptides were identified within the capsid-coding region of the EMCV polyprotein, which supports the notion that EVs released by EMCV-Wt infected cells incorporate mature virions. In contrast, no viral peptides were detected in EMCV-L$^{Zn}$ induced EVs. Hence, the presence of the Leader strongly promotes the packaging and/or release of EV-enclosed EMCV.

**The EMCV Leader promotes the release of specific virus-containing EV subsets.** The reduction in EV-enclosed virus released by cells infected with the Leader-deficient virus prompted us to further explore the role of the Leader in the induction of EV release. Western blot analysis of various known EV-associated proteins indicated that EMCV-Wt infection induced an increase in the release of both 10 and 100 K EVs, while EMCV-L$^{Zn}$ infection did not boost the release of 10 or 100 K EVs (Fig. 2a). Using an in-house developed, high-resolution flow cytometry-based approach[36,37] with which the quantity and quality of EVs can be assessed at the single-particle level, we confirmed that the

population of EVs released from EMCV-Wt-infected cells comprised both 10 and 100 K EVs, while EMCV-L$^{Zn}$ mainly consisted of 100 K EVs, similar to mock EVs (Fig. 2b). We therefore further focused on comparing 100 K EVs released during EMCV-Wt and EMCV-L$^{Zn}$ infection. First, using high-resolution flow cytometry, we compared the number of EVs released by non-infected and infected cells at different timepoints post-infection. We observed that EMCV-Wt infection caused a sharp increase in the overall number of 100 K EVs released from 6 h p.i. onwards (Fig. 2c). However, in cells infected with EMCV-L$^{Zn}$, this virus-induced increase in EV release was completely abolished, resulting in a significantly lower number of released EVs at 8 h p.i. for EMCV-L$^{Zn}$ infected cells compared to EMCV-Wt infected cells (Fig. 2c, d). A closer examination of the flow cytometry dot plots revealed an additional phenotypic difference between the released EVs. EMCV-Wt induced the release of two EV subpopulations with different forward scattered light (FSC) intensities (Fig. 2e), of which the population of FSC-high EVs is most efficient in spreading the infection to other cells[5]. Over the course of the infection, an increase in the percentage of FSC-high EVs released by cells infected with EMCV-Wt was observed (Fig. 2f, g). In contrast, the percentage of FSC-high EVs released from cells infected with EMCV-L$^{Zn}$ remained comparable to mock cells throughout the infection. These results indicate that the EMCV Leader protein promotes the release of a specific subset of FSC-high EVs that were previously shown to efficiently contribute to EMCV transmission.

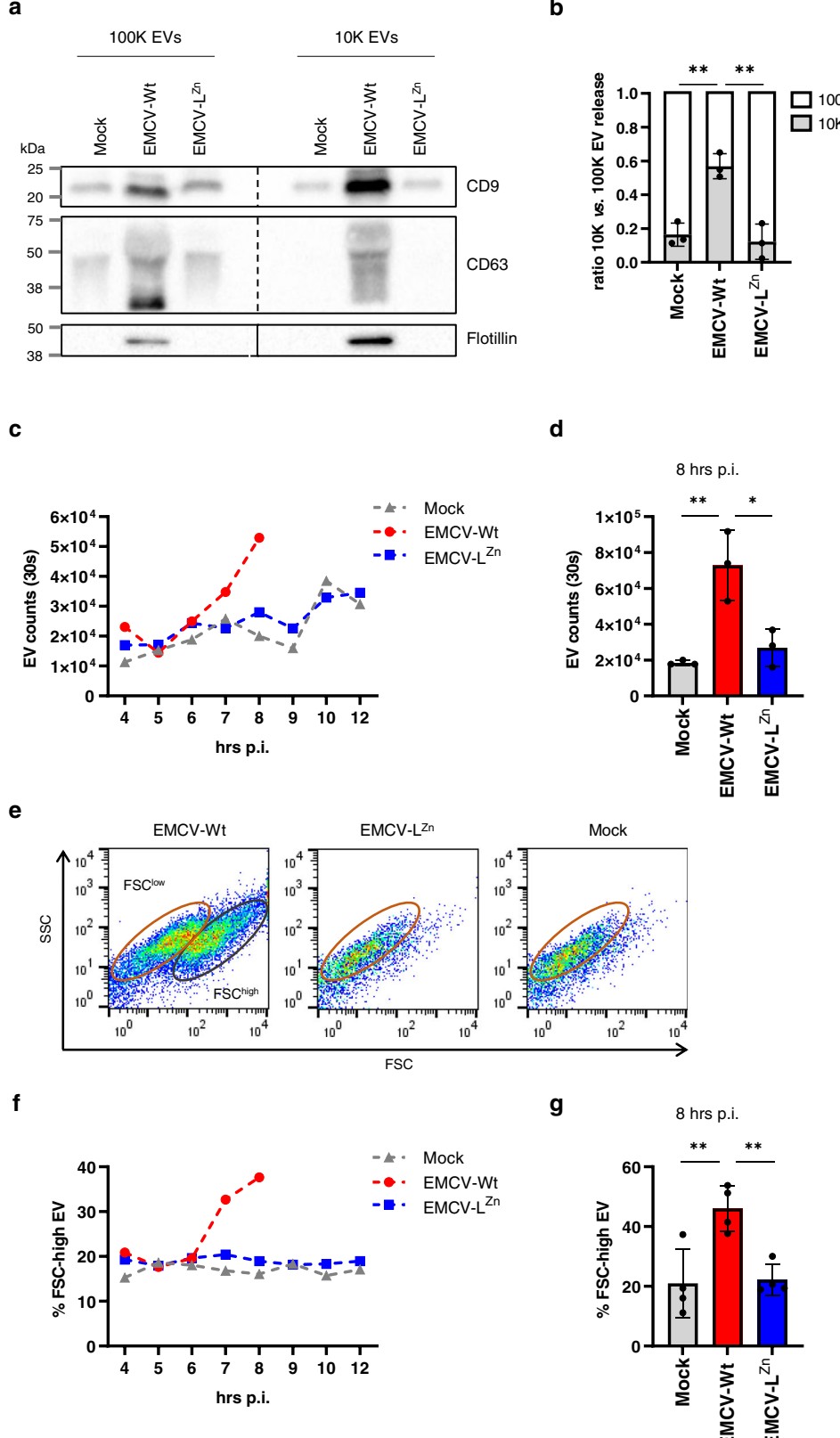

**The EV proteome suggests a potential role for secretory autophagy in Leader-induced EV release**. To further investigate how the Leader protein affects the composition and release of EVs and EV-enclosed virus, we compared the host protein composition of EV preparations ($n = 3$ replicates) from cells infected with EMCV-Wt or EMCV-L$^{Zn}$. Proteins involved in EV formation

and release have frequently been identified as integral components of EVs. Comparative analysis of EV proteomes can therefore provide valuable information on how the Leader protein affects pathways involved in EV formation and release. Of the total number of proteins identified in EVs from infected cells by LC-MS/MS ($N = 234$ for EMCV-Wt, and $N = 275$ for EMCV-

**Fig. 2 The EMCV Leader is required for the induction of an infectious EV subpopulation with an FSC-high phenotype. a** 10,000×$g$ (10 K) and 100,000×$g$ (100 K) ultracentrifugation pellets were collected from an equal number of EV-producing cells 8 h after mock, EMCV-Wt, or EMCV-L$^{Zn}$ infection and analyzed by western blotting for the presence of EV-marker proteins CD9, CD63, and flotillin. Depicted is a representative of $n = 3$ independent experiments. **b–g** EVs were isolated from cell culture supernatants of mock, EMCV-Wt, and EMCV-L$^{Zn}$ infected cells by differential ultracentrifugation and density gradient centrifugation at indicated timepoints p.i. EVs were fluorescently labeled with CFSE and analyzed by high-resolution flow cytometry. **b** The contribution of 10 K vs. 100 K EVs to the overall EV release by mock, EMCV-Wt, and EMCV-L$^{Zn}$ infected cells was quantified. Mean values ± SD are presented for $n = 3$ independent experiments. **left $p = 0.0025$, **right $p = 0.0015$. **c, d** Depicted is the increase over time of 100 K EV numbers present in 1.06–1.10 g/ml fractions measured in a fixed time window of 30 s (**c**) and the number of EVs present in those density fractions at 8 h p.i. (**d**). ** $p = 0.0047$, *$p = 0.0107$. **e** Representative FSC/SSC dot plots are depicted of the samples in (**d**). Gates delineate FSC-high and FSC-low EVs. **f, g** Presented are the percentage of FSC-high EVs of the total EVs present in the 1.08 g/ml density fraction at the indicated timepoints (**f**) and the percentage of FSC-high EVs at 8 h p.i. (**g**). ** left $p = 0.0061$, ** right $p = 0.0084$. For (**c**) and (**f**) a representative image of $n = 2$ individual experiments is shown. For (**d**) and (**g**) mean values ± SD are presented for $n = 4$ independent experiments. $p$ values were assessed by one-way ANOVA with Tukey's multiple comparisons test. Source data are provided as a Source Data file.

L$^{Zn}$), 190 proteins were detected for both viruses, while 44 proteins were exclusively identified in EMCV-Wt-induced EVs (Fig. 3a and Supplementary Table 1). The 44 proteins that were dependent on the Leader for incorporation into EVs were subjected to functional enrichment analysis. In the Gene Ontology (GO) domain "cellular component", these proteins were highly enriched (>10-fold enrichment vs. genome) for ER-Golgi intermediate compartment (ERGIC) membrane, and secretory granule membrane components, as well as components of ER-to-Golgi transport vesicles (Fig. 3b). Subsequent analysis of protein interaction networks revealed an interconnected group of proteins in EMCV-Wt EVs that were linked to these enriched GO-terms (Fig. 3c). Furthermore, GO analysis for "biological process" revealed highly enriched terms associated with vesicular transport between ER and Golgi, as well as post-Golgi vesicle-mediated transport (Fig. 3d). Protein-network analysis revealed the involvement of a core group of interacting proteins, which included regulators of trafficking between ER and Golgi (TFG, TMED9, and TMED10), proteins implicated in trafficking between Golgi and endosomes (Rab14 and LMAN2), and SNARE proteins (Sec22b, SNAP23, and STX4) (Fig. 3e). The plasma membrane-resident SNARE proteins SNAP23 and STX4 were recently implicated in EV release by regulating the fusion of MVBs with the plasma membrane[38,39]. Moreover, by interacting with Sec22b upon its recruitment to autophagosomal membranes, this combination of SNARE proteins was also shown to facilitate a secretory arm of the autophagy pathway that enables the release of the pro-inflammatory cytokine IL-1β[40]. Both ERGIC and Golgi compartments are considered membrane sources for autophagosome formation[41–43]. Enrichment of GO-terms related to these compartments in EMCV-Wt EVs could support the notion that autophagosomes intersect with the site of EV biogenesis.

The intersection of autophagosomes with compartments involved in EV biogenesis can result in the extracellular release of autophagic cargo, such as LC3[44–47]. Therefore, we expanded our analysis of the secretome of infected cells by western blotting for the autophagy marker LC3 in crude ultracentrifugation pellets. We observed that the ultracentrifugation pellets from EMCV-Wt infected cell supernatant contained LC3, whereas there was no notable release of LC3 from cells infected with EMCV-L$^{Zn}$ (Fig. 3f). These data support the role of the EMCV Leader in the activation of a secretory arm of the autophagy pathway. Part of the LC3 released by EMCV-Wt-infected cells was sensitive to treatment with proteinase K (Fig. 3g), indicating that this LC3 was not enclosed within the EV lumen, but instead was secreted alongside or associated to EVs. However, part of the LC3 signal, likely corresponding to the lipidated form of LC3 (LC3II), was protected from protease digestion in the absence of detergents, indicating that LC3 is also partially incorporated in EMCV-Wt induced EVs. In support of these data, we observed

that the lipidated form of LC3 co-segregated with EV-marker proteins in density gradients, while part of the LC3 was not associated with EVs and remained in the high-density bottom fractions of the gradients (Fig. 3h). The Leader-dependent secretion of LC3, combined with the presence of Sec22b, SNAP23, and STX4 as well as Golgi/ERGIC components in EVs from EMCV-infected cells hints at a role for secretory autophagy in the release of virus-containing EVs.

**The EMCV Leader modulates autophagic flux during infection.** To further investigate the role of the EMCV Leader in the activation of secretory autophagy, we first assessed whether the Leader affected the overall level of autophagy in infected cells. Immunoblot analysis of EMCV-Wt and -L$^{Zn}$ infected cells showed that both viruses promoted the conversion of LC3I to its phosphatidylethanolamine-conjugated form (LC3II), indicating an increase in the number of autophagosomes compared to uninfected cells (Fig. 4a). However, this increase in LC3II/I ratios was more pronounced in cells infected with EMCV-Wt. In addition, we noted that infection with EMCV-Wt, but not EMCV-L$^{Zn}$, consistently increased the presence of an additional, intermediate LC3 band. This banding pattern has previously been ascribed to a pro-form of LC3, although the migration patterns of LC3 forms remain a matter of debate[48–50]. We next assessed whether the Leader-induced increase in LC3II-containing autophagosomes was the result of a defect in the degradation of autophagosomes by fusion to lysosomes. Hereto, we made use of cells expressing tandem-fluorochrome tagged LC3 (mCherry-EGFP-LC3) in which autophagosomes can be distinguished from autolysosomes based on their EGFP:mCherry fluorescence ratio (Fig. 4b). In this reporter system, the drop in pH caused by the fusion of autophagosomes with late endosomes or lysosomes causes the loss of LC3-EGFP fluorescence in autolysosomes while the more acid-resistant mCherry fluorescence persists. Using confocal imaging, we detected an increase in the overall surface area of autophagosomes (EGFP$^+$mCherry$^+$ puncta) versus autolysosomes (EGFP$^-$mCherry$^+$ puncta), constituting either an increase in their number or size, over the course of EMCV-Wt infection (Fig. 4c). In contrast, no differences were observed in the ratio between these two compartments in EMCV-L$^{Zn}$ infected cells compared to mock-treated controls, although an increase in LC3II levels was also observed upon EMCV-L$^{Zn}$ infection. We investigated whether the accumulation of autophagosome membranes was caused by a Leader-induced defect in the lysosomal system, and hence an inability to form acidified autolysosomes, by tracing the intensities of lysotracker stained cells. Lysotracker staining of acidified (endo)lysosomal compartments was similar between cells infected with EMCV-Wt and EMCV-L$^{Zn}$ (Fig. 4d). Thus, the Leader induces an accumulation of autophagosomes without causing defects in lysosomal function.

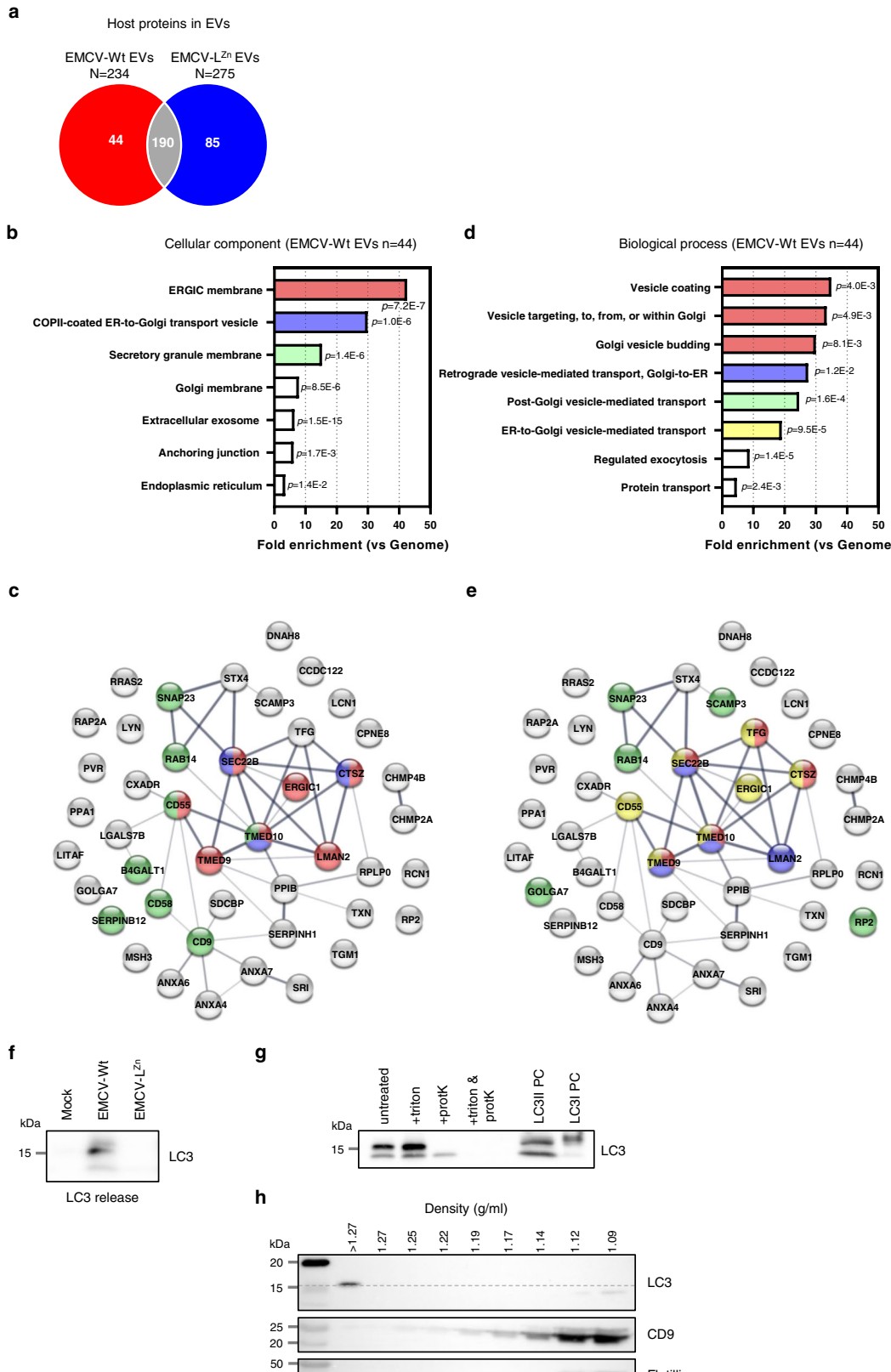

Next, we determined whether the observed reduction in autophagosome turnover in the presence of the Leader contributed to the induction of secretory autophagy and the release of EV-enclosed virus. To this end, we tested the impact of rapamycin treatment on EMCV-Wt infected cells. Rapamycin is an inhibitor of mTOR and a well-known stimulator of autophagic flux. Treatment with rapamycin partially restored the balance in the ratio of autophagosomes vs. autolysosomes without affecting virus replication or cell viability (Fig. 5a and Supplementary Fig. 3), indicating that infected cells were still capable of autolysosome formation. Despite an increase in autophagic degradation, we observed no concomitant reduction in the

**Fig. 3 Analysis of the EV proteome supports a Leader-dependent role for secretory autophagy in the formation of virus-induced EVs.** EVs released by cells infected with either EMCV-Wt or EMCV-L$^{Zn}$ were isolated and EV-protein composition was analyzed by LC-MS/MS. **a** Venn diagram depicting the number of unique and overlapping host proteins identified in at least 2 out of 3 replicate preparations of EMCV-Wt and EMCV-L$^{Zn}$ EVs. **b–e** Functional enrichment analysis for the 44 unique proteins in EMCV-Wt EVs was combined with an evaluation of protein–protein interactions using the STRING database. **b, d** Functional annotations are listed for the GO domains *cellular component* (**b**) and *biological process* (**d**). Bars depict the fold enrichment for the indicated GO-terms compared to the genome, with concomitant *p* values as assessed by one-tailed Fisher's exact test with Bonferroni correction for multiple testing. **c, e** Proteins in interactome networks are indicated by their gene name and thickness of the connecting lines correlates with the strength of data support for interaction. Color-coding of individual proteins corresponds to their association with GO-terms that are enriched >10-fold compared to the genome for *cellular component* (**c**) or *biological process* (**e**). **f** Ultracentrifugation pellets (100 K) of cell culture supernatants from an equal number of mock, EMCV-Wt, or EMCV-L$^{Zn}$ infected cells were analyzed by western blotting for the presence of extracellular LC3. **g** EMCV-Wt samples from (**f**) were treated with proteinase K in the absence or presence of 0.2% triton and analyzed by western blotting for the presence of LC3. LC3I PC and LC3II PC are positive control samples for western blot detection of LC3I and LC3II, respectively. **h** EMCV-Wt samples from (**f**) were floated into a sucrose density gradient. Individual gradient fractions were analyzed for the presence of LC3 and the EV markers CD9 and flotillin by western blotting. Representative images of *n* = 3 independent experiments are shown for (**f, g**) and *n* = 2 independent experiments for (**h**). Source data are provided as a Source Data file.

release of LC3 upon rapamycin treatment, indicating that the level of secretory autophagy was not diminished (Fig. 5b). Moreover, rapamycin had no effect on the overall amount of EV-enclosed virus released by infected cells (Fig. 5c), although the altered cellular condition induced by this drug caused the virus to be preferentially packaged in 10 K rather than 100 K EVs (Fig. 5d). Similar results were obtained with the autophagy-inducing drug amiodarone, which also had no effect on LC3 release from EMCV-Wt infected cells or the release of EV-enclosed virus (Supplementary Fig. 4). These data suggest that Leader-induced secretion of autophagosome cargo and EV-enclosed viruses is not caused by defective or insufficient (auto) lysosomal degradation. Instead, we propose that the Leader specifically induces the activation of a secretory arm of the autophagy pathway.

**The induction of secretory autophagy promotes EV-enclosed virus release.** Finally, we investigated whether the induction of secretory autophagy and the packaging of EMCV virions in EVs were causally linked. If that were the case, we reasoned that pharmacological induction of secretory autophagy might rescue the defect in EV-enclosed virus release caused by mutation of the EMCV Leader. To test this hypothesis, we treated cells with apilimod prior to infection with EMCV-L$^{Zn}$. Apilimod prevents lysosome regeneration and prolonged exposure of cells was previously shown to trigger the extracellular release of autophagosome cargo[51]. To verify that the effects induced by apilimod were due to the specific induction of secretory autophagy rather than the inhibition of lysosome function during the course of the infection, we treated cells with bafilomycin as a negative control. Bafilomycin functions as an inhibitor of lysosomal degradation, similar to apilimod, by preventing lysosomal acidification (Fig. 4d). Under our treatment regimes, neither drug negatively affected cell viability (Supplementary Fig. 5). Apilimod, but not bafilomycin, efficiently induced secretory autophagy in EMCV-L$^{Zn}$ infected cells, indicated by the robust presence of LC3 in 100 K EV pellets (Fig. 6a). However, both drugs induced a similar increase in total EV release (Fig. 6b). Importantly, apilimod treatment (partially) rescued EV-enclosed virus release by EMCV-L$^{Zn}$ infected cells (Fig. 6c), despite causing an overall decrease in intracellular virus production (Fig. 6d). In contrast, bafilomycin treatment did not promote the capacity of EMCV-L$^{Zn}$ infected cells to release EV-enclosed viruses. While increasing EV-enclosed virus release, the EVs induced by apilimod treatment did not display the FSC-high phenotype observed for EVs induced by EMCV-Wt virus infection (Supplementary Fig. 6), suggesting that apilimod treatment does not mimic all effects of a competent Leader protein. Together, these findings indicate that the release of viruses in EVs is not merely controlled by the

overall amount of EV production by infected cells. Instead, the packaging of virus in EVs could be specifically enhanced by the pharmacological induction of secretory autophagy, a process that is enabled by the viral Leader protein during EMCV infection

## Discussion

The data presented here highlight the first example of a non-structural viral protein regulating the release of EV-enclosed viruses. We showed that the EMCV Leader is crucial in triggering the release of EV-enclosed progeny virus from EMCV-infected cells. We demonstrated that the Leader triggers the release of a specific subset of EVs previously demonstrated to efficiently transmit EMCV infection[5]. In contrast, EV released by cells infected with a Leader-deficient virus was comparable to mock-infected cells, and these constitutively released EVs failed to efficiently package virus particles. Finally, we demonstrate the role of secretory autophagy in the Leader-mediated induction of EV-enclosed virus release.

Our findings represent a novel addition to the variety of functions ascribed to the EMCV Leader and shed new light on the role of this protein in disease. The Leader is of critical importance for virus pathogenesis in vivo, and its absence strongly reduces morbidity and mortality in mice[31]. The contribution of the Cardiovirus Leader to efficient virus replication and spreading has been ascribed to its ability to interfere with the induction of antiviral defense pathways, such as the type I IFN response, the cellular stress response, and the induction of apoptotic cell death[30–32,52]. By promoting the release of EV-enclosed viruses, the Leader additionally enables the formation of infectious particles that are functionally distinct from naked virions, both in their uptake dynamics and in their accessibility to the humoral immune system due to the surrounding EV membrane[1,3,10,13,18,19,53]. As a result, the formation of EV-enclosed virus represents an additional mechanism via which the Leader may enable immune escape and efficient virus spread during EMCV infection.

Our data indicate that the Leader induces the packaging and release of viruses in specific EV subpopulations and that the secretory arm of autophagy plays a role in the formation of these virus-carrying EVs. Autophagy was previously shown to be required for the unconventional secretion of cytosolic proteins that lack a signal peptide. Importantly, many of the proteins shown to depend on autophagy for extracellular release were also found to be packaged in EVs. The cytokine IL-1β, a prototypical target of secretory autophagy, was found to be partially shed within exosomes and microvesicles[54,55]. Other examples include annexin A2, the chaperon α-Crystallin B, and the high mobility group box 1 protein, which were shown to depend on the functionality of the autophagic machinery as well as the exosome biogenesis pathway for release[46,56–59]. In most cases, a correlation

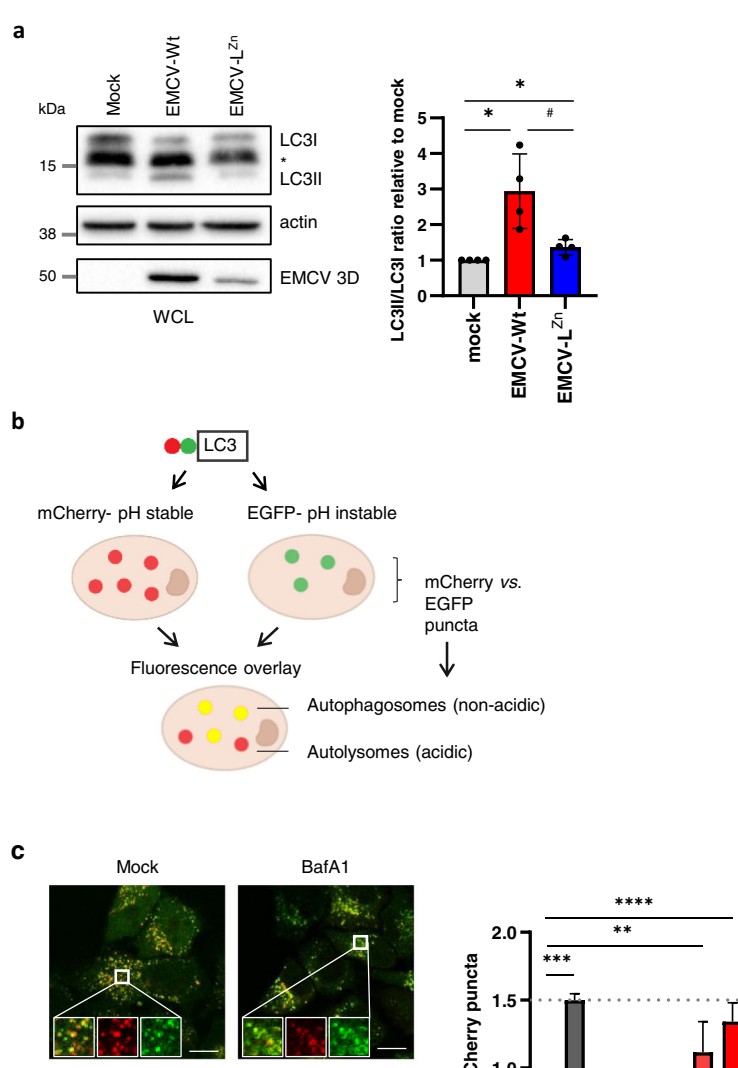

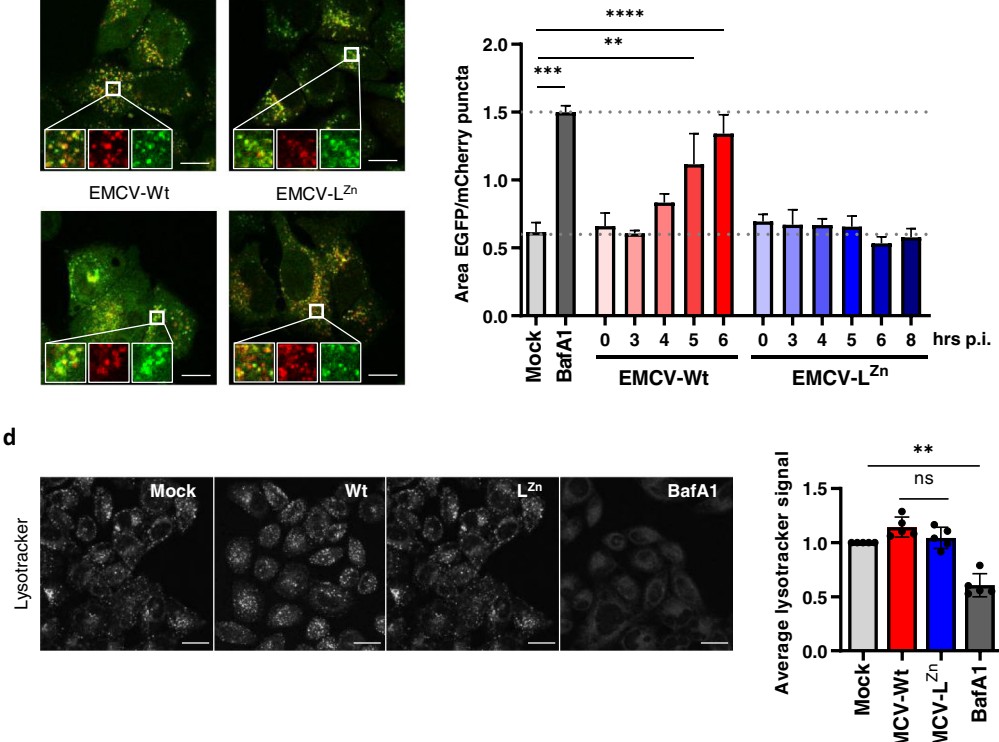

with the induction of LC3 secretion was demonstrated, highlighting the involvement of a secretory arm of the autophagy pathway in the packaging of cargo molecules in EV. Similarly, we observed a correlation between the extracellular release of LC3 and the enhanced packaging of EMCV in EVs during infection.

Moreover, pharmacological induction of autophagic secretion supported the EV-mediated release of virions.

Three models have been proposed to explain how secretory autophagy could contribute to the release of EV-enclosed cargo molecules. Firstly, autophagosomes may fuse directly with the

**Fig. 4 The EMCV Leader promotes the accumulation of autophagosomes during infection. a** Mock-infected cells and cells infected with EMCV-Wt or EMCV-L$^{Zn}$ were harvested 8 h p.i. Whole-cell lysates were analyzed for conversion of LC3I to LC3II by western blotting and the LC3II:LC3I ratios were calculated relative to mock-infected cells. *Indicates a detected LC3 band with intermediate size. Actin was included as a loading control and EMCV-3D to confirm infection. Depicted are representative blots and the mean ± SD of $n = 4$ independent experiments. *$p = 0.0341$ (bottom), $p = 0.0257$ (top) as determined by a two-tailed one-sample $t$-test. # $p = 0.0437$ using a two-tailed $t$-test. **b** Schematic overview of the mCherry-EGFP-LC3 autophagy reporter system. Using spot detection on LC3 reporter cells imaged by confocal microscopy, the accumulation of LC3 in autophagosomal vs. autolysosomal compartments can be monitored. Whereas autophagosomes appear as mCherry+ EGFP+ LC3+ puncta, autolysosomes are detected as mCherry+ EGFP− LC3+ puncta due to a loss of EGFP fluorescence in the acidic environment of the autolysosome lumen. **c** LC3 reporter cells were infected with EMCV-Wt or EMCV-L$^{Zn}$ for the indicated time or treated with bafA1 for 6 h and imaged for EGFP and mCherry fluorescence using confocal microscopy. Some EGFP+ compartments seem to lack mCherry signal, but this is likely due to the lower detection efficiency of mCherry versus EGFP. Bar graphs display a quantification of the ratio between the total surface area of EGFP and mCherry positive puncta as a measure for the ratio between the autophagosomes and autolysosomes present in the cells. Indicated are mean values ± SEM of ≥7 separate images with 2–5 cells each from a representative of $n = 3$ individual experiments. **$p = 0.0083$, ***$p = 0.0001$, ****$p < 0.0001$ as assessed by one-way ANOVA with Dunnett's multiple comparison test. Scale bar = 20 μm. (**d**) Infected cells were stained with 50 nM lysotracker and compared to mock-infected cells or bafA1 treated controls 6 h p.i. using live-cell confocal imaging. Depicted are representative images showing lysotracker fluorescence staining. Bar graphs display the average cellular staining intensity relative to uninfected controls. Indicated are mean values ± SD of five independent experiments. Scale bar = 30 μm. **$p = 0.0012$, as assessed by a one-sample $t$-test. ns, $p = 0.1400$ as assessed by an unpaired, two-tailed $t$-test. Source data are provided as a Source Data file.

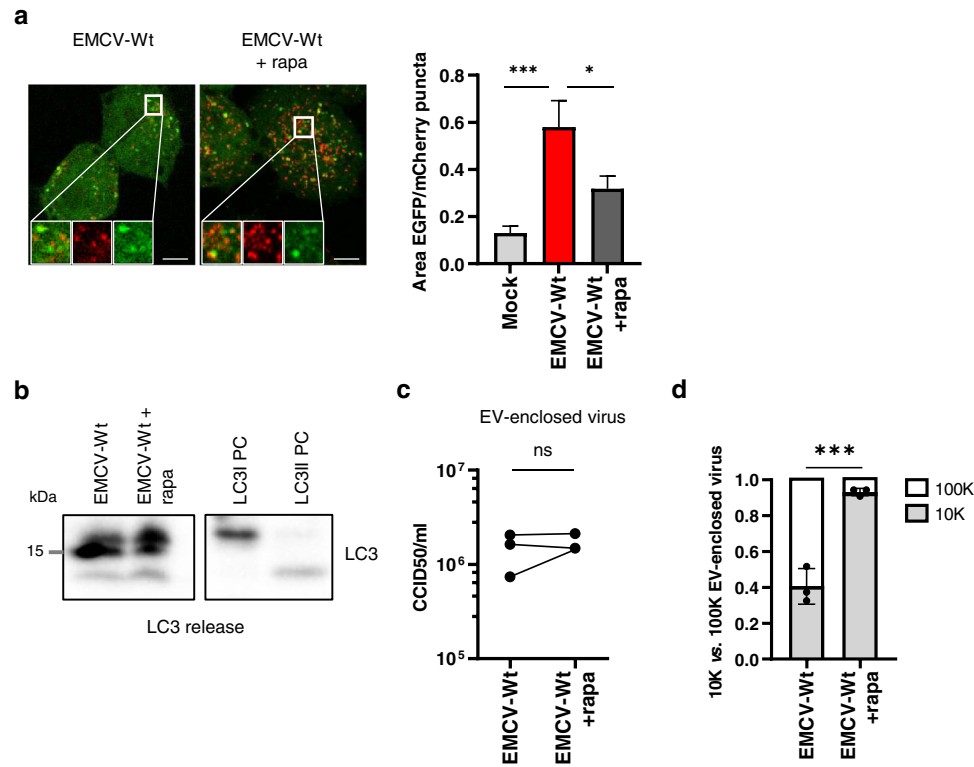

**Fig. 5 Enhancing autophagic degradation does not inhibit Leader-induced secretory autophagy and EV-enclosed virus release.** EMCV-Wt infected cells were treated with 200 nM rapamycin 1 h p.i. onwards to increase autophagic flux and enhance autolysosome formation. **a** Representative confocal microscopy images of infected mCherry-EGFP-LC3 reporter cells 6 h p.i. Bar graphs display the ratio between the total surface area of EGFP and mCherry positive puncta, that correspond to autophagosomes (EGFP+ mCherry+) and autolysosomes (EGFP−mCherry+). Indicated are mean values ± SEM of ≥7 separate images with 2–5 cells each from a representative of $n = 3$ individual experiments. ***$p = 0.0008$, *$p = 0.0434$ as assessed by one-way ANOVA with Tukey's multiple comparison test. Scale bar = 10 μm. **b** Western blot analysis of LC3 release in ultracentrifugation pellets harvested 8 h p.i. from cell supernatants of an equal number of cells. Depicted is a representative image of $n = 3$ independent experiments. LC3I PC and LC3II PC are positive control samples for western blot detection of LC3I and LC3II, respectively. **c, d** 10 K and 100 K EVs were isolated and purified using density gradients. Depicted is the total amount of EV-associated infectivity (**c**) and its distribution over 100 K vs. 10 K EVs (**d**) (mean ± SD of $n = 3$ independent experiments) as determined by end-point dilution assay. ***$p = 0.0009$, ns $p = 0.6599$ as assessed by an unpaired, two-tailed $t$-test. Source data are provided as a Source Data file.

plasma membrane to release the autophagosome's inner membranous vesicle. Secondly, autophagosomes may fuse with multivesicular bodies (MVBs) or endosomes to result in hybrid compartments, referred to as amphisomes, that are subsequently released[29]. Finally, the autophagic machinery may even be directly recruited to endosomes to aid in EV cargo selection at the MVB limiting membrane, bypassing the need for autophagosome formation altogether[44,45]. In the case of EMCV infection, we observed that the majority of the LC3 released from cells was accessible to degradation by proteinase K. This is compatible with

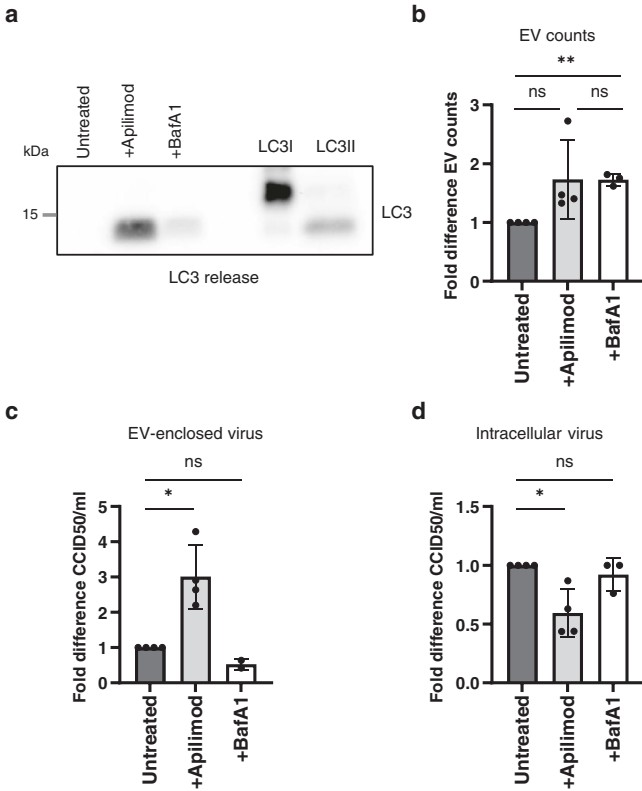

**Fig. 6 Apilimod reactivates secretory autophagy and promotes EV-enclosed virus release in the absence of the EMCV Leader.** **a** EMCV-L$^{Zn}$ infected cells were treated with 500 nM apilimod for 16 h prior to infection or 200 nM BafA1 1 h p.i. onwards. EV-containing 100,000×*g* ultracentrifugation pellets were isolated from supernatants of an equal number of cells at 8 h p.i. and analyzed by western blot for the presence of extracellular released LC3. Depicted is a representative image of *n* = 2 independent experiments. LC3I PC and LC3II PC are positive control samples for western blot detection of LC3I and LC3II, respectively. **b** 100 K EVs were labeled with CFSE, purified by density gradient centrifugation, and analyzed by high-resolution flow cytometry. Bars indicate the mean fold increase in EV release relative to untreated controls ±SD from *n* = 3 (BafA1) or *n* = 4 (apilimod) independent experiments. **p = 0.0064, ns (left) *p* = 0.1148 as assessed by a two-tailed one-sample *t*-test, or two-tailed *t*-test for ns (right) *p* = 0.9936. **c, d** Depicted is the mean fold increase in EV-enclosed infectivity (**c**) and intracellular virus titers (**d**) relative to untreated controls ±SD corresponding to the samples in (**b**) as determined by end-point dilution assay. For **c** *p* = 0.0210, ns *p* = 0.1464, for **d** *p* = 0.0285, ns *p* = 0.4226, as assessed by a two-tailed one-sample *t*-test. Source data are provided as a Source Data file.

partial disassembly of autophagic membranes in amphisomes, which may liberate LC3 from its membrane confinement and result in the release of LC3 that is exposed to the environment. Conversely, the release of intact autophagosomes as well as the direct recruitment of LC3 to the outer MVB membrane is expected to lead to the selective release of membrane-enclosed LC3. In support of a potential role for amphisomes in the formation of EV-enclosed viruses during infection, we found that EVs from EMCV-infected cells were enriched in proteins of both Golgi/ERGIC and endosomal compartments. ERGIC and Golgi have been implicated as important membrane sources for autophagosome formation[41–43]. Structural rearrangements and dispersal of the Golgi and ERGIC were previously reported in EMCV-infected cells, indicating potential recruitment of these membranes to other sites[60]. Therefore, we speculate that viral disturbance in the Golgi/ERGIC in presence of the EMCV Leader

induces the formation of ERGIC-derived autophagosomal membranes that fuse with endosomal compartments, where they become incorporated into budding EV. Endosomal proteins enriched in EVs from EMCV-infected cells included the Lipopolysaccharide-induced tumor necrosis factor-alpha factor (LITAF) and the ESCRTIII complex protein charged multi-vesicular body protein 2 A (CHMP2A). LITAF and CHMP2A have been implicated in both MVB formation and autophagy, and CHMP2A was previously also recognized to play an important role in the EV-mediated release of the picornavirus HAV[61–63]. Our study provides leads on how virions may be packaged in EVs, but the exact mechanism via which the autophagosomes and endosomal systems intersect and facilitate this process remains to be determined.

Limited knowledge is currently available on the molecular mechanisms that mediate the rerouting of autophagosomes onto a pathway for extracellular release. It has been proposed that specific regulators may be recruited to autophagosomes that enable their preferential fusion with the PM or other non-lysosomal compartments, such as GRASP55 and TRIM16[40,55,64,65]. For the picornavirus CVB3, we recently identified a role for the transcription factor EB (TFEB), a known regulator of lysosomal exocytosis, in the activation of secretory autophagy[66]. TFEB depletion limited the ability of CVB3 to induce LC3 release during infection, which correlated with an overall decrease in EV-enclosed virus release. In contrast, other studies have suggested that autophagic secretion may be a response to defects in autophagosome clearance. Inhibitors of lysosomal function, such as apilimod, were shown to trigger the release of autophagic cargo upon prolonged exposure[51,67]. Examples in pathological settings include the defects in autophagosome clearance that accompany the accumulation of prion-like proteins such as α-synuclein, which leads to their export from cells together with LC3[67–72]. Several viruses are also known to interfere with lysosomal function. In specific, picornaviruses belonging to the enterovirus genus have been reported to disrupt the molecular machinery necessary for autophagosome-lysosome fusion[73–75]. However, our data indicate that the EMCV Leader protein enabled the secretion of autophagosomal cargo without interfering with lysosomal degradation or causing an inherent block in autophagosome-lysosome fusion. Despite an intracellular accumulation of autophagosomes, no lysosomal defects were observed in infected cells. In addition, autolysosome formation in EMCV-infected cells could still be enhanced by rapamycin treatment. Hence, the induction of EV-enclosed and autophagic cargo release by EMCV does not appear to be a response to a limitation in cellular degradation capacity caused by the presence of the Leader. Rather, we propose that virus infection leads to specific activation of this secretory pathway. Given the diverse effects of picornavirus species on autophagosome-lysosome fusion, picornaviruses may employ different mechanisms to prevent degradation and enable the secretion of EV-enclosed virions sequestered by the autophagic machinery.

The class of picornavirus security proteins, to which the EMCV Leader belongs, have evolved strongly overlapping functions in targeting host cell processes, despite showing little to no sequence or structural similarities[30]. Follow-up studies should therefore address whether the role of the EMCV Leader in the induction of virus-carrying EV that we discovered here also represents a repeating feature among the non-structural viral proteins of this exceptionally large group of viruses. Ultimately, the identification of the viral and host factors involved in the regulation of EV-enclosed release will shed new light on the role of these particles in the viral life cycle and increase our understanding of the intricate relationship between autophagy, EVs, and virus–host interactions.

## Methods

**Cells and virus**. Human cervical carcinoma cells (HeLa R19) were a kind gift from Dr G. Belov (University of Maryland, USA) and baby hamster kidney cells (BHK21, ATTC CCL-10) were obtained from the American Type Culture Collection (Rockville, MD). HeLa R19 cells stably expressing mCherry-EGFP-LC3 were obtained through stable transfection of plasmid DNA (Addgene #22418), followed by puromycin selection and generation of monoclonal cell populations by limited dilution. Cell lines were cultured in a humidified incubator at 37 °C and 5% $CO_2$ in Iscove's modified Dulbecco's medium (IMDM; Lonza, Basel, Switzerland) with 2 mM Ultraglutamine (Lonza), or Dulbecco's Modified Eagle Medium (high glucose, GlutaMAX$^{TM}$, Thermo Fischer Scientific, Waltham, MA, USA), supplemented with 10% fetal calf serum (FCS; GE Healthcare Bio-Sciences, Chicago, IL), 100 U/mL penicillin and 100 µg/mL streptomycin (Gibco, Paisley, UK).

Both EMCV-Wt and EMCV-L$^{Zn}$ stocks were obtained by transfection of BHK21 cells with in vitro RNA transcripts of the previously described infectious cDNA clone pM16.1, which contains a copy of the EMCV genome with a shortened poly-C tract[76]. The Zn-finger mutation in the Leader was introduced by site-directed mutagenesis as described previously[31]. Virus-containing culture supernatant was harvested after observing virus-induced cytopathogenic effect (CPE) and subjected to three consecutive freeze-thaw cycles. Virus was concentrated from cell-free culture supernatant by high-speed ultracentrifugation through a 30% sucrose cushion at 80,000×g for 16 h in an SW32 rotor (k-factor 321) (Beckman Coulter, Brea, CA, USA).

**Drug treatments**. To block lysosomal acidification and subsequent autophagosomal degradation, cells were treated with 200 nM bafilomycin A1 (Cayman Chemical, Ann Arbor, MI) from 1 h p.i. onwards or for the indicated duration of time. To enhance autophagic flux, cells were treated with 300 nM rapamycin (Tocris Bioscience, Bristol, UK) or 5 µM amiodarone hydrochloride (Tocris) from 1 h p.i. onwards. To activate secretory autophagy, cells were pretreated with 500 nM apilimod (AchemBlock, Burlingame, CA) for 16 h, followed by infection in the presence of apilimod for the indicated duration of time.

**Isolation of EVs**. For isolation of EVs from pre-lytic EMCV-infected cells, HeLa R19 cells were seeded in T75 culture flasks and infected the next day at MOI = 10. At 1 h p.i., the unbound virus was removed by washing the cells three times with PBS, and cells were provided with a culture medium containing EV-depleted FCS. To remove EVs from FCS, 30% FCS in IMDM was ultracentrifuged for 16–20 h at 100,000×g in an SW32 rotor (k-factor 256.8) and passed through a 0.22 µm filter. Cell culture supernatants were collected after indicated culture durations and were sequentially centrifuged at 2× 200×g for 10 min, 2× 500×g for 10 min to remove cell debris. 10 K EVs were isolated by ultracentrifugation at 10,000×g for 30 min, and 100 K EVs by subsequent centrifugation of the 10,000×g supernatant at 100,000×g for 65 min in an SW40 rotor (k-factor 280.3) (Beckman Coulter). For analysis of LC3 release in EV-bound and non-EV bound form, 500×g culture supernatants were directly pelleted at 100,000×g. EV-enriched ultracentrifugation pellets were resuspended in 20 µL PBS containing 0.2% BSA (cleared from aggregates by ultracentrifugation for 16–20 h at 100,000×g). For further purification, resuspended EV pellets were mixed with 60% iodixanol (Optiprep; Axis-Shield, Oslo, Norway) to a final concentration of 45% iodixanol and overlaid with a linear gradient of 40–5% iodixanol in PBS. Density gradients were centrifuged at 192,000×g for 15–18 h in an SW40 rotor (k-factor 144.5). Gradient fractions of 1 mL were collected and densities were assessed by refractometry. Unless otherwise indicated, fractions within the 1.04–1.10 g/ml density range were pooled for the characterization of EVs and EV-enclosed virus release based on the buoyant densities validated previously for naked EMCV virions and EVs[5]. For western blot analysis directly from gradient fractions, EV pellets were mixed 1:5 with a 2.5 M sucrose solution and overlaid with a linear gradient of 2.0–0.4 M sucrose in PBS. Density gradients were centrifuged at 192,000×g for 14 h in an SW60 rotor (k-factor 45), followed by the collection of gradient fractions of 0.33 ml.

**Confocal microscopy**. HeLa R19 LC3 reporter cells expressing the mCherry-EGFP-LC3 plasmid were infected with EMCV-Wt or EMCV-L$^{Zn}$ at MOI = 10 for confocal imaging. At the indicated timepoints, cells were fixed in 4% PFA. Cells were imaged using a Leica SPEII/DMI4000 confocal microscope with hardware settings to image EGFP and mCherry channels sequentially by 488 and 561 nm laser excitation using recommended emission spectral filter settings. The acquisition was performed using an ACS APO 63x oil immersion objective (N.A. 1.3) with the pinhole set to 1 airy disk. Data were analyzed using ImageJ software (1.52i)[77]. For each condition ≥7 images were taken with 2–5 cells each. Background subtraction was performed using Gaussian blur (sigma 7) and EGFP and mCherry positive puncta were segmented by thresholding. Summed surface area measurements of EGFP and mCherry positive puncta per image field were collected and ratios of these measurements are depicted.

For lysotracker staining, HeLa R19 cells were seeded in FluoroBrite$^{TM}$ Dulbecco's modified Eagle medium (DMEM; Thermo Fischer Scientific), supplemented with 10% FCS, 1% P/S, and 2 mM ultraglutamine. Cells were infected at MOI = 25 or treated with 200 nM BafA1 and mounted on a Nikon A1R confocal microscope while maintaining cell culture conditions (37 °C, 5% $CO_2$) in a tabletop humidified

culture control unit (TOKAI Hit). Live-cell imaging of 4 × 4 fields of view was performed using a 60x CFI PLAN APO oil objective or a 20x PLAN APO VC air objective with widened pinhole settings (20.43 um) ~6 h p.i. following the addition of 50 nM LysoTracker Red DND-99 (Thermo Fischer Scientific). For quantification of the average lysotracker staining intensity of the cells, images were processed using the NIS elements 5.1 general analysis module (Nikon Microsystems). A rolling ball background correction (6 µm) and three iterations of smoothing were performed before setting a low-intensity threshold to segment cells and exclude extracellular areas from analysis (monitored by DIC images and the lysotracker staining). A higher threshold setting was determined using the BafA1 treated controls to exclude segmented areas consisting of cells that had undergone virus-induced lytic cell death (supported by DIC images).

For dsRNA staining, HeLa R19 cells were infected at MOI = 10 and fixed using 4% PFA 6–8 h p.i. Cells were permeabilized in PBS + 0.1% Triton X-100 for 10 min, washed and incubated in block buffer (PBS + 2% BSA + 50 mM $NH_4Cl$) for 30 min. Permeabilized samples were stained for 45 min with primary mouse-α-dsRNA (1:200-1:1000, J2; English Scientific & Consulting, Hungary) followed by a second incubation step with DAPI plus donkey-α-mouse Alexa-488 or Alexa-647 (1:200, Invitrogen, UK). All antibodies were diluted in block buffer and between antibody incubations, samples were washed three times in block buffer. Finally, the coverslips were washed 1x in block buffer and 1x in MiliQ before being mounted in FluorSafe (Calbiochem, San Diego, CA) or ProLong$^{TM}$ Diamond Antifade Mountant (Thermo Fischer Scientific). Samples were imaged using a 60x or 100x CFI PLAN APO oil objective on a Nikon A1R confocal microscope with hardware settings to image the relevant channels sequentially using recommended emission spectral filter settings. Images were processed using ImageJ software and scored manually using 2–3 images per sample containing 11–42 cells per image.

**Flow cytometric analysis of cells**. Cell viability was assessed using 7-AAD viability staining (eBioscience, San Diego, CA) or Fixable Viability Dye eFluor$^{TM}$ 506 (eBioscience, San Diego, CA) according to the manufacturer's protocols. In short, adherent cells were harvested by trypsinization and pooled with any detached cells recovered from the supernatant following centrifugation at 200×g for 10 min. For 7-AAD staining cells were washed with PBS, and stained with 5 µL dye per 1 × 10^6 cells for 5 min at RT. Cells were analyzed using a BD FACS Canto II (BD Biosciences, San Jose, CA) with BD FACS Diva software (version 6.1.3). For the Fixable Viability Dye eFluor$^{TM}$ 506 labeling, cells were washed twice in ice-cold PBS followed by staining on ice for 30 min using a 1:1000 working dilution. Unbound dye was removed by washing with PBS and the cells were fixed in 1% PFA. Cells were analyzed using a CytoFLEX LX (Beckman Coulter). Data analysis was performed using FlowJo v10.07 software (FlowJo LLC, Ashland, OR). Heat shocked cells were taken along in all experiments as a positive control. To this end, cells were incubated for 3 min at 65 °C followed by immediate placement on ice for 1 min.

**High-resolution flow cytometric analysis of EVs**. For high-resolution flow cytometric analysis of EVs, 100 K EVs were labeled with 30 µM CFSE (Invitrogen, Carlsbad, CA) in 50 µl PBS for 1 h at RT. Unbound CFSE was separated from EVs by density gradient centrifugation as described above. EVs in gradient fractions were fixed with 2% PFA for 30 min and diluted in PBS for high-resolution flow cytometric analysis on a BD Influx flow cytometer with optimized configuration, as previously reported[36,37]. In short, thresholding was applied on fluorescence generated by CFSE-labeled EVs passing the 488 nm laser. Fluorescent 100 and 200 nm polystyrene beads (FluoSpheres, Invitrogen) were used to calibrate the fluorescence, and forward (FSC) and side (SSC) scattered light settings. Samples were measured at low pressure (sheath fluid: 5 PSI, sample: 4.2 PSI) using a 140 µm nozzle with event rates below 10,000 per second. Measurements were acquired in a fixed time window of 30 s to allow direct comparison of EV concentrations in parallel samples. Data analysis was performed using FlowJo software or FCS expression v3 (De Novo Software, Los Angeles, CA).

**End-point dilution**. Intracellular infectivity levels were evaluated in samples of infected HeLa R19 cells that were subjected to three consecutive freeze-thaw cycles, followed by centrifugation for 10 min at 500×g to remove cell debris. Extracellular infectivity levels in the pre-lytic stage of infection were determined in supernatants of infected cells after a similar pre-clarification step. Infectivity in purified EVs was assessed by direct sampling from EV-containing density gradient fractions. BHK21 cells in 96-well clusters were infected with three to fivefold serial dilutions of the material described above, and virus titers expressed in Median Cell Culture Infectious Dose (CCID50 values) were calculated 3 days after infection using the Spearman–Karber calculation method.

**SDS-PAGE and western blotting**. For analysis of cell lysates, cells were lysed in RIPA buffer (40 mM Tris-HCl pH 8, 0.5% sodium deoxycholate, 1% Triton X-100, 150 mM sodium chloride, 0.1% sodium dodecyl sulfate) with a protease inhibitor cocktail (Roche). Lysates were cleared by centrifugation at 16,000×g for 15 min and protein concentration was determined using a Pierce BCA assay kit (Thermo Scientific, Waltham, MA) according to the manufacturer's protocol. Cell lysates, ultracentrifugation pellets containing EVs, and density gradient fractions were denatured at 100 °C for 4 min in non-reducing Laemmli sample buffer (LSB:

62.5 mM Tris-HCl pH 6.8, 2% SDS, 10% glycerol) for detection of CD9, CD63, and flotillin, and reducing LSB (+20 mM 2-Mercaptoethanol) for detection of LC3, EMCV-3D, and actin. Proteins were separated on 12.5% sodium dodecyl sulfate-polyacrylamide gels by electrophoresis (SDS-PAGE) and transferred to Immobilon 0.20 or 0.45 µm pore-sized PVDF membranes (Merck Millipore Ltd., Cork, Ireland) by wet transfer. Membranes were incubated with blocking buffer (0.2% fish skin gelatin (FSG; Sigma-Aldrich) (CD9, flotillin, CD63, LC3) or 4% BSA (LC3, actin, EMCV 3D) + 0.1% Tween-20 in PBS) for 1 h, and for >16 h at 4 °C with the following primary antibodies: mouse-α-CD63 (1:1000, clone TS63; Abcam, Cambridge, UK), mouse-α-CD9 (1:2000, clone HI9a; Biolegend, San Diego, CA), mouse-α-Flotillin-1 (1:1000, clone 18/Flotillin-1; BD Biosciences), mouse-α-LC3 (1:1000, clone 18/Flotillin-1; BD Biosciences), mouse-α-LC3 (1:500, clone 5F10; 1:50, clone 2G6; Nanotools, Teningen, Germany), rabbit-α-LC3 (1:1000, polyclonal; MBL International, Woburn, MA), mouse-α-actin (1:30,000, clone AC-15, Sigma-Aldrich), and mouse-α-EMCV 3D polymerase (1:100, clone 3B7, sc-65633, Santa Cruz Biotechnology, Dallas, TX) diluted in blocking buffer. Membranes were subsequently washed with blocking buffer and incubated for 1 h with HRP-coupled goat-α-mouse secondary antibody (1:10,000; Jackson ImmunoResearch Laboratories Inc., West Grove, PA) or goat-α-rabbit secondary antibody (1:2500, P0448, DAKO, Denmark) diluted in blocking buffer. Prior to imaging, membranes were washed ≥3x with PBS-0.1% Tween-20 followed by PBS. ECL solution (SuperSignal West Dura Extended Duration Substrate, Thermo Scientific) was used for detection on a Bio-Rad ChemiDoc imager and images were analyzed by Image Lab software (Bio-Rad).

**Proteinase K digestion.** 100 K EV pellets isolated from the supernatants of EMCV-Wt infected cells were resuspended in PBS and incubated in the absence or presence of 0.2% Triton X-100 for 15 min at RT. Then, 45 ng/µl Proteinase K (Roche, Basel, Switzerland) was added followed by a second incubation step at 37 °C for 30 min. Proteinase K was inactivated by the addition of 1 mM PMSF (Roche) and samples were mixed with LSB for SDS-PAGE and western blot analysis.

**Proteomics sample preparation.** For MS analysis, EV-containing density fractions (1.06–1.13 g/ml) were diluted in PBS containing 0.1% BSA and centrifuged at 100,000×g for 90 min in an SW32 rotor. EV pellets were resuspended in reducing Laemmli sample buffer and denatured at 100 °C for 4 min. The protein content of EV preparations was estimated by SDS-PAGE followed by SYPRO® Ruby protein staining densitometry. In short, sample aliquots were used for protein separation on 8–16% TGX-Criterion gels (Bio-Rad, Hercules, CA). After electrophoresis, gels were fixed in 40% methanol/10% acetic acid and stained with SYPRO® Ruby (Invitrogen, Carlsbad, CA), followed by destaining in 10% methanol/6% acetic acid. Gels were imaged on a Bio-Rad ChemiDoc imaging system and densitometry quantitation was performed using Image Lab software (Bio-Rad). For LC-MS/MS analysis, EV-protein input was equalized between conditions and proteins were separated by SDS-PAGE on 8–16% TGX-Criterion gels. Gels were fixed in 50% methanol/10% acetic acid, stained with Coomassie Brilliant Blue R-250 (Bio-Rad), and destained sequentially with 40% methanol/10% acetic acid and Milli-Q. In-gel digestion using trypsin (Promega, Madison, WI) was performed as previously described[78]. After extraction with 100% acetonitrile the samples were dried and reconstituted in 10% formic acid/5% DMSO.

**High-resolution LC-MS/MS and data analysis.** The samples were analyzed with a Q-Exactive (Thermo Scientific, Bremen, Germany) connected to an Agilent 1290 Infinity LC system, a trap column of 20 mm × 100 µm ID Reprosil C18 (Dr. Maisch, Ammerbuch, Germany), and a 450 mm × 75 µm ID Poroshell C18 analytical column (Zorbax, Agilent Technologies, Santa Clara, CA), all packed in-house. Solvent A consisted of 0.1 M acetic acid (Merck, Kenilworth, NJ) in Milli-Q (Millipore, Burlington, MA), while solvent B consisted of 0.1 M acetic acid in 80% acetonitrile (Biosolve, Valkenswaard, The Netherlands). Trapping was performed at a flow rate of 5 µl/min for 10 min and peptides were eluted at 100 nl/min for 55 min. The mass spectrometer was operated in data-dependent mode to automatically switch between MS and MS/MS. For the MS/MS analysis the 15 most intense ions in the survey scan (375 to 1600 m/z, resolution 60,000, AGC target 3e6) were subjected to HCD fragmentation (resolution 30,000, AGC target 1e5), with the normalized collision energy set to 27% for HCD. The signal threshold for triggering an MS/MS event was set to 500 counts. The low mass cut-off for HCD was set to 180 m/z. Charge state screening was enabled, and precursors with an unknown charge state or a charge state of 1, 6–8, and >8 were excluded. Dynamic exclusion was enabled (exclusion size list 500, exclusion duration 8 s). MS data processing and analysis MS raw data were processed with Proteome Discoverer (version 1.4.1.14, Thermo Fisher Scientific). Peptide identification was performed using Mascot 2.3 (Matrix Science, UK). Peak lists were generated from the raw data files using Proteome Discoverer version 1.4.1.14 (Thermo Fisher Scientific) and searched against the current (Consulted in July 2018) UniProtKB/Swiss-Prot database (all entries for identification of EMCV proteins, Homo sapiens for identification of host proteins), supplemented with frequently observed contaminants. Trypsin was chosen as the enzyme and two missed cleavages were allowed. Carbamidomethylation was set as a fixed modification and oxidation was set as a variable modification. The searches were performed using a peptide mass tolerance of 50 ppm and a product ion tolerance of 0.05 Da, followed by data filtering using a percolator, resulting in a 1% false discovery rate (FDR). Only ranked 1 PSMs with Mascot scores >20 were accepted. Functional enrichment analysis was performed for cellular components and biological processes using PANTHER (via http://geneontology.org). By hierarchical sorting of analysis results, only the most specific subclasses of GO-terms are presented, while redundant parental terms are excluded. Enriched GO-terms were ranked by fold enrichment versus the genome. Only significantly enriched GO-terms are presented. P values are determined by Fisher's exact test with Bonferroni correction for multiple testing. Protein–protein interactions were analyzed using the STRING database version 11.0 (https://string-db.org/), with the minimum required interaction score set to medium confidence (0.400).

**Statistics.** Graphs were generated and statistically analyzed as indicated in the figure legends using Graphpad Prism version 8/9 (Graphpad Software, CA).

**Reporting summary.** Further information on research design is available in the Nature Research Reporting Summary linked to this article.

## Data availability
The raw mass spectrometry proteomics data generated in this study have been deposited to the ProteomeXchange Consortium via the PRIDE partner repository with the dataset identifier PXD033906[79]. We have submitted all relevant data of our experiments to the EV-TRACK knowledgebase (EV-TRACK ID: EV220089, https://evtrack.org/search.php)[80]. Source data are provided with this paper.

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

## Acknowledgements

We are grateful to dr. Ger Arkesteijn for help with high-resolution flow cytometric analysis, as well as the Center for Cell Imaging Utrecht for allowing access to their facility for confocal imaging and training. We thank dr. Carla Ribeiro for providing antibodies against LC3 and Dr. Guillaume van Niel for helpful discussions. This work was supported by The Netherlands Organization for Scientific Research (NWO-ALW grant number ALWOP.351) (https://www.nwo.nl/en) as well as the European Research Council under the European Union's Seventh Framework Program [FP/2007-2013] / ERC Grant Agreement number [337581] to E.N.M.N.tH. (https://erc.europa.eu/funding/starting-grants). This research was part of the Netherlands X-omics Initiative and partially funded by NWO (Project 184.034.019 to M.A.). The funders had no role in study design, data collection, and analysis, decision to publish, or preparation of the manuscript.

## Author contributions

S.G.v.d.G., K.A.Y.D., F.J.M.v.K., and E.N.M.N.tH. conceived the project. S.G.v.d.G., K.A.Y.D., F.J.M.v.K., and E.N.M.N.tH. wrote the manuscript. S.G.v.d.G. and K.A.Y.D. designed and performed experiments. S.S.G. and M.A. performed mass spectrometry and M.J.C.v.H performed subsequent data analysis. H.H.R. performed confocal imaging of LC3 reporter cells. R.W.W. together with K.A.Y.D. performed and analyzed lysotracker staining. E.N.M.N.tH and F.J.M.v.K. supervised the project.

## Competing interests

The authors declare no competing interests.
