## [Peer Review File · Nature Communications]

Peer review comments, first round of review

Reviewer #1 (Remarks to the Author):

This manuscript demonstrates that the leader protein of the encephalomyocarditis virus can promote the virions to be released inside of the extracellular vesicles. This is an interesting study that emphasizes the egress strategy of RNA virus. The results showed that leader protein plays an important role in stimulating the release of EVs and virions enclosed EVs. The authors proposed that the leader protein promotes the release of virions enclosed EVs through affecting autophagy pathway. The experiments were well designed and described with details. Although the PLoS Pathogen paper published in 2019 by the same group reveals extracellular vesicles released by EMCV infected cells are infectious and Robinson et al., demonstrated that LC3 protein could be detected in virus-containing extracellular vesicles, this is the first paper showed that viral protein can trigger the autophagy to promote the release of EV-enclosed virus.

Comments:

1. Line 120, " Virus packaging in EVs is believed to be the main form of virus release in the early or pre-lytic stages of the infection" please provide the reference.
2. Please provide the data to prove that virus released in the early stage of infection is mainly packaging inside of EVs.
3. Whether the mutant virus can cause the death of the host cells?
4. Whether EMCV-Wt and EMCV-Lzn can infect host cells with the similar efficiency?
5. Whether the viral protein could be detected in EVs released from EMCV-Wt and EMCV-Lzn-infected cells?
6. In Figure 3F, the presence of exosome should be confirmed by using multiple antibodies. For example, antibodies against CD9, TSG101, and calnexin can be applied.
7. In Figure 4A and 5B, the expression of viral protein should be examined in Western blot to confirm the infection.
8. Except autophagy induction, rapamycin has been reported to be able to inhibit protein kinase C and micropinocytosis. Thus, the results observed may not come from the ability of rapamycin in suppressing autophagy. To avoid this, more than one reagents should be used. In addition to rapamycin, there are other reagents such as simvastatin and amiodarone that have been known for their ability in triggering autophagy through different strategies.
9. Since autophagy induction can induce the release of virions, is it possible that suppression of autophagosome formation can result in the inhibition of virus release?
10. In figure 6A, the expression of β actin or GAPDH should be performed.
11. Please check the label in Figure 6A.
12. How to tell the differences between the EVs (without virus) and EV-enclosed virus?
13. What is the percentage of virions that are released within EVs among the total released virus?
14. Previous study suggests that more than one virions could be enclosed in one extracellular vesicle, which could contribute to the enhanced infectivity of EV-enclosed virus. Have you ever check this phenomenon in your EVs?

15. The results showed in this manuscript suggest that EMCV leader plays an important role in promoting the release of virions inside the extracellular vesicles. However, the effects of overexpression of leader protein on viral egress have not been examined. For example, plasmids encoding leader sequence can be transfected into EMCV-Lzn cells to observe whether the EV-enclosed virus release could be restored.

Reviewer #2 (Remarks to the Author):

The manuscript entitled "The encephalomyocarditis virus Leader promotes the release of virions inside extracellular vesicles via the induction of secretory autophagy" by van der Grein et al. explores the role of Leader protein in release of EV-enclosed viruses from encephalomyocarditis virus (EMCV) infected cells. Previous studies by the group and others showed that picornaviruses are released from infected cells via packaging into extracellular vesicles that can modulate infection efficiency, that these EVs contain autophagy-regulatory protein LC3 and that several picornaviruses actively induce autophagy to support virus replication and release. Van der Grein et al. have built on that and showed here that EMCV induce release of EV-enclosed viruses from infected cells by activating the secretory arm of autophagy through the Leader protein activity. Thereby they provide the first evidence for a crucial role of non-structural viral protein in the non-lytic release of picornaviruses via packaging in EVs.

The study is an important contribution to the field of picornaviruses (and other naked viruses) as it connects viral (EMCV Leader) protein and the cell process of secretory autophagy to non-lytic release of EV-enclosed EMCV. Additionally it contributes to the field of extracellular vesicle as it explores the role of secretory autophagy in the process of EV biogenesis/release.

The conclusions of the study are well-supported by the presented data and meaningfully put into broader context of the secretory autophagy-mediated release of (naked virions via) EVs. Sufficient context and references were provided. A minor suggestion would be to refrain from stating the role of picornavirus security proteins is explored in the study, as only EMCV Leader protein is studied and the data is not discussed in relation to evolutionary conservation of the Leader protein function (or to other security proteins) in picornaviruses.

The study is well-designed, analyzed and presented, and authors use state-of-the-art techniques (e.g. high resolution flow cytometry) to support study conclusions. Still, what is missing in the study is uncovering the mechanism of the observed effect of the Leader protein on secretory autophagy. Is it a direct or indirect effect? Is it connected to other known effects of the Leader dependent on the Zn-finger motif, like well-conserved anti-interferon activity (IFN1 is a known inducer of autophagy), for which mechanism was described?

Below suggested minor improvements would further strengthen the work:

- Figure 1: Please include definition of TCID50 in the Materials and methods section.
- Figure 2: The size interval of EV population detected and analyzed by the high resolution flow cytometry is not mentioned but would be important for comparisons to outcomes of other studies (additionally, why were EVs defined as particles 50-300 nm in size, as generally the sizes go up to 1000 nm?).
- Figure 3: Was LC3 detected by mass spectrometry? Why were studies of EV-bound LC3 with western blot performed on crude 100K pellets and not EV 1.06-1.1 g/ml density fraction, which represents EVs devoid of naked EMCV virions?
- Figure 4: Ratio between autophagosomes vs. autolysosomes is evaluated based on total surface area of EGFP and mCherry positive puncta, therefore I would advise not to interpret this as increase in autophagosomes numbers as increase in total surface area of puncta can be due to increase in number or size of autophagosomes. Additionally, just as a matter of interest, what represents the only green fluorescence observed in some cells; is the mCherry-EGFP-LCS plasmid polycistronic?
- Figure 5: Treatment of EMCV-wt infected cells with rapamycin (stimulator of autophagic flux) re-routed the packaging of viruses in the 10K rather than 100K EVs. What is authors explanation and do these EVs also bind/contain LC3?

- Figure 6: Apilimod was shown to induce EV-enclosed virus release in the absence of EMCV Leader activity, but how does it effect the release of FSChigh population, representing infectious EV subpopulation?

Reviewer #3 (Remarks to the Author):

The work of Van der Grein et al describes a novel role of the EMCV Leader protein in the release of virus enclosed in EVs by a mechanism involving the induction of secretory autophagy. The work is original and relevant to the fields of virology and EVs. The conclusions are supported by the data. The methodology is sound and presented in sufficient detail to be reproducible.

However, I have some comments;

1) Although the experimental settings are explained in enough detail to allow reproducibility, I encourage the authors to submit the data to the EVTRACK website (<http://www.evtrack.org>).

2) In line 125, the authors state that cells infected with EMCV-Lzn mutant do not release 10K EVs. The authors did not show these data. However, I consider it important to show how EV release is changed in cells infected with the mutant virus compared to the control. Characterization of the 10k and 100K EVs in terms of numbers of particles and/or EV markers will be useful to understand how the Leader is changing virus secretion within EVs. Showing differences in infectivity in these fractions and/or their behavior after optiprep separation, although not necessary, would be of added value to having a complete picture of the effects of the Leader on virus secretion inside EVs.

3) Western blots showing 100K EVs in general, or at least the western blot in figure 3A in particular, should show different EV markers according to the guidelines of the International Society for EVs (MISEV2018# 30637094). The Western blots in figures 3G, 5B, and 6A do not contain kDa annotation. The Western blots in figures 3F, 3G and 6A are only representative images of two experiments. Ideally, these westerns should be performed in at least three independent experiments, and their quantification could be provided. The information about the number of repetitions done for figures 5B and 4A is missing.

4) EV count in Figures 2b and 6b was performed by high-resolution flow cytometry after labeling of the EVs with CFSE. However, this labeling may not label/sufficiently label all EVs subtypes. It would be an added value to include another way of counting EVs, for example using an NTA-based technique.

5) There is an error in the numbering of Sup Fig 3 in line 264

We appreciate the constructive feedback and advice from the reviewers to further improve our manuscript. In the pages that follow, we provide a point-by-point response to the reviewers' comments. Revisions in the new version of the manuscript are indicated in red font.

REVIEWER COMMENTS

Reviewer #1 (Remarks to the Author):

This manuscript demonstrates that the leader protein of the encephalomyocarditis virus can promote the virions to be released inside of the extracellular vesicles. This is an interesting study that emphasizes the egress strategy of RNA virus. The results showed that leader protein plays an important role in stimulating the release of EVs and virions enclosed EVs. The authors proposed that the leader protein promotes the release of virions enclosed EVs through affecting autophagy pathway. The experiments were well designed and described with details. Although the PLoS Pathogen paper published in 2019 by the same group reveals extracellular vesicles released by EMCV infected cells are infectious and Robinson et al., demonstrated that LC3 protein could be detected in virus-containing extracellular vesicles, this is the first paper showed that viral protein can trigger the autophagy to promote the release of EV-enclosed virus.

Comments:

1. Line 120, " Virus packaging in EVs is believed to be the main form of virus release in the early or pre-lytic stages of the infection" please provide the reference.

Authors' response:

We further clarified the statement about this point in line 123-125 of the revised manuscript. In recent years, various research groups have reported virus release via EVs during the early stages of infection, prior to lysis-mediated massive release of naked virions. To substantiate this point, we added references to a number of literature review papers that discuss the occurrence and importance of this form of virus release.

2. Please provide the data to prove that virus released in the early stage of infection is mainly packaging inside of EVs.

Authors' response:

Our findings indicate that EMCV release via EVs mainly occurs in the earlier stages of infection, and precedes the massive release of naked virus in the lytic stage. Ratios between naked and EV-enclosed virus in the pre-lytic stage of EMCV infection have been provided in our previous publication¹. As also mentioned under point 1, we clarified this in lines 123-125 of the revised manuscript.

3. Whether the mutant virus can cause the death of the host cells?

Authors' response:

It is well-documented in literature that wildtype EMCV causes necrotic cell death and cell lysis in *in vitro* cultures because the Leader protein counteracts apoptotic cell death². In contrast, cells infected with mutant viruses with an inactive Leader protein are reported to undergo apoptosis². In our experiments we observed morphological signs of apoptosis, such as cell shrinking and membrane blebbing, in EMCV-L^{Zn} infected cells after ≥ 12 hours of culture. Extending the culture

beyond this timepoint resulted in apoptotic cell death of the EMCV-L^{Zn} infected cells. In our revised manuscript we have included additional statements on cell death in EMCV-L^{Zn} infected cells in lines 120-121.

4. Whether EMCV-Wt and EMCV-Lzn can infect host cells with the similar efficiency?

Authors' response:

In Figure 1A of our manuscript we show that under the experimental conditions used in our study (single round of infection, high MOI) EMCV-Wt and EMCV-L^{Zn} have similar replication kinetics and that the difference in intracellular production of infectious virus is substantially smaller than the difference in the release of extracellular virus. We agree with the reviewer that information on the percentage of infected cells for EMCV-Wt and EMCV-L^{Zn} is of added value. In our revised manuscript, we therefore added these data in the new Supplementary figure 1. The data clearly show that the percentage of infected cells is comparable for EMCV-Wt and EMCV-L^{Zn}.

5. Whether the viral protein could be detected in EVs released from EMCV-Wt and EMCV-Lzn-infected cells?

Authors' response:

To specifically address the question whether viral proteins are present in EVs released by EMCV-Wt and EMCV-L^{Zn} infected cells, we performed LC-MS/MS analysis of EV proteomes and mapped the peptide hits against the EMCV polyprotein. We detected 17 different unique viral peptides in our EMCV-Wt EV preparations, of which most were recognized to be viral capsid components. In contrast, no viral peptides were identified in EMCV-L^{Zn} EVs. These results are presented in Figure 1D of our revised manuscript. In agreement with these findings, we show in Figure 1C that EMCV-Wt EVs, but not EMCV-L^{Zn} EVs, contain infectious virus. Combined, these data demonstrate that mature virions, consisting of infectious genomic RNA coated by a protein capsid, are present in EVs released by EMCV-Wt infected cells, while EVs from EMCV-L^{Zn} infected cells are devoid of viral particles.

6. In Figure 3F, the presence of exosome should be confirmed by using multiple antibodies. For example, antibodies against CD9, TSG101, and calnexin can be applied.

Authors' response:

As requested by the reviewer, we have included western blot analysis of EV marker proteins CD9, CD63, and Flotillin-1 (new Figure 2A in revised manuscript).

7. In Figure 4A and 5B, the expression of viral protein should be examined in Western blot to confirm the infection.

Authors' response:

In Figure 4A, we have now included western blot analysis for the presence of the EMCV non-structural protein 3D in cellular extracts, to confirm that infection is established in both EMCV-Wt and EMCV-L^{Zn} infected cells. This is in agreement with the data of our standard, CCID50-based, assessment of intracellular virus production, shown in Figure 1A for the EMCV-Wt versus EMCV-L^{Zn} conditions. Confirmation of infection elsewhere in the manuscript (Figure 6D, and Supplementary figures 3A and 4A) has been performed using this CCID50-based method.

We have not added detection of viral protein to Figure 5B, which shows western blot analysis of EVs instead of cells. The presence or absence of viral proteins in EVs cannot be used to assess cellular infection efficiency, because enhanced infection does not necessarily lead to increased incorporation of viral proteins in EVs.

8. Except autophagy induction, rapamycin has been reported to be able to inhibit protein kinase C and micropinocytosis. Thus, the results observed may not come from the ability of rapamycin in suppressing autophagy. To avoid this, more than one reagents should be used. In addition to rapamycin, there are other reagents such as simvastatin and amiodarone that have been known for their ability in triggering autophagy through different strategies.

Authors' response:

We appreciate the reviewer's constructive feedback on this point. Indeed rapamycin not only triggers autophagy, but can also influence cell survival, cell growth, and cell proliferation through its inhibitory effect on the pleiotropic kinase mTOR. In our revised manuscript, we have added experimental data on the effect of amiodarone on the extracellular release of LC3 and the release of EV-enclosed virus (new Supplementary figure 4). Like rapamycin, amiodarone can induce autophagic flux via inhibition of mTOR, but also promotes autophagy in a mTOR-independent, calpain-dependent manner^{3,4,5}. Our data with amiodarone corroborate our findings with rapamycin; while both drugs enhanced autophagy in EMCV-Wt infected cells, neither drug prevented the infection-induced LC3 secretion and EV-enclosed virus release. A description of these data can be found in lines 260-262 of the revised manuscript.

9. Since autophagy induction can induce the release of virions, is it possible that suppression of autophagosome formation can result in the inhibition of virus release?

Authors' response:

Treatment of cells with inhibitors that limit the early stages of autophagy and knockdown of autophagy-related genes that regulate autophagosome formation have been reported to reduce picornavirus titers^{6,7,8}. However, Mauthe *et al* (JCB 2016) demonstrated that inhibition of autophagy induction severely compromises replication of several viruses including EMCV⁹. Thus, it is difficult to assess whether inhibition of autophagosome formation specifically affects virus release, because any observed effect may be a direct consequence of the reduced intracellular virus titers. The strength of our approach is that we assess the interplay between EV-virus release and late stages of autophagy (autophagic flux and secretory autophagy), using interference strategies (mutant viruses and pharmacological inhibition) that do not, or hardly, affect virus replication (as confirmed in Figures 1A and 6D, and Supplementary figures 3A and 4A).

10. In figure 6A, the expression of b actin or GAPDH should be performed.

Authors' response:

The cytoplasmic proteins β -Actin and GAPDH are routinely used as loading control in western blot analysis of cellular extracts. However, for western blot analysis of EVs, β -Actin and GAPDH cannot be used for normalization, since the distribution of these proteins in EVs is unknown and may not be stable under different experimental conditions. As a general strategy for western blot analysis of EVs (Figures 2A, 3F-3H, 5B, and 6A) we collected EVs from a fixed number of cells for each experimental condition. Observed differences in protein levels can therefore result from changes in the protein

composition of EVs, or in the number of released EVs. We added a more detailed description of this strategy in the relevant figure legends.

11. Please check the label in Figure 6A.

Authors' response:

Figure 6A in the manuscript has been adapted to include kDa annotation.

12. How to tell the differences between the Evs (without virus) and EV-enclosed virus?

Authors' response:

No biochemical or biophysical characteristics have yet been identified based on which EVs that do or do not carry virus can be separated. Electron microscopy has been used by other groups to demonstrate that virus-containing EVs exist. Using a high-throughput flow cytometric approach, we previously demonstrated that EVs with high and low capacity to transfer infection to new cells differ in light scattering properties¹. Ongoing studies in our lab aim to delineate whether the increased light scattering properties signify the presence of virus particles or other biophysical/biochemical properties of EVs that promote infection. In our current manuscript, this question is less relevant, because we study the efficiency with which the total population of released EVs transfers the infection to new cells.

13. What is the percentage of virions that are released within EVs among the total released virus?

Authors' response:

As stated under point 2, assessment of the amount of naked versus EV-enclosed virus in the pre-lytic stage of EMCV infection was provided in our previous publication¹. In our current manuscript, we exclusively focus on EV-enclosed virus.

14. Previous study suggests that more than one virions could be enclosed in one extracellular vesicle, which could contribute to the enhanced infectivity of EV-enclosed virus. Have you ever check this phenomenon in your EVs?

Authors' response:

Indeed, it is possible that multiple virions are packaged per EV. This may facilitate en-bloc delivery to target cells and possibly also 'genetic cooperativity' between viral quasi-species, thereby increasing the infectivity. Whether there is variation in the number of virions per EV for EVs released by EMCV-infected cells is beyond the scope of the current manuscript, but will be addressed in future experiments.

15. The results showed in this manuscript suggest that EMCV leader plays an important role in promoting the release of virions inside the extracellular vesicles. However, the effects of overexpression of leader protein on viral egress have not been examined. For example, plasmids encoding leader sequence can be transfected into EMCV-Lzn cells to observe whether the EV-enclosed virus release could be restored.

Authors' response:

It is indeed expected that reconstitution of EMCV-L^{Zn} infected cells with functional Leader protein could restore EV-enclosed virus release. However, this requires a very precise dosing and timing of the expression of the Leader protein so that the levels of this protein match the EMCV-L^{Zn} infection. Besides the fact that this is very difficult to control, the overexpression of Leader protein is also highly toxic to cells (unpublished data from our lab). We consider the comparison between Leader-competent wildtype viruses and mutant viruses with an inactive Leader protein to be the best possible strategy to study Leader function. This is an ideal approach to study Leader function in the physiologically relevant context of a viral infection.

Reviewer #2 (Remarks to the Author):

The manuscript entitled "The encephalomyocarditis virus Leader promotes the release of virions inside extracellular vesicles via the induction of secretory autophagy" by van der Grein et al. explores the role of Leader protein in release of EV-enclosed viruses from encephalomyocarditis virus (EMCV) infected cells. Previous studies by the group and others showed that picornaviruses are released from infected cells via packaging into extracellular vesicles that can modulate infection efficiency, that these EVs contain autophagy-regulatory protein LC3 and that several picornaviruses actively induce autophagy to support virus replication and release. Van der Grein et al. have built on that and showed here that EMCV induce release of EV-enclosed viruses from infected cells by activating the secretory arm of autophagy through the Leader protein activity. Thereby they provide the first evidence for a crucial role of non-structural viral protein in the non-lytic release of picornaviruses via packaging in EVs.

The study is an important contribution to the field of picornaviruses (and other naked viruses) as it connects viral (EMCV Leader) protein and the cell process of secretory autophagy to non-lytic release of EV-enclosed EMCV. Additionally it contributes to the field of extracellular vesicle as it explores the role of secretory autophagy in the process of EV biogenesis/release.

The conclusions of the study are well-supported by the presented data and meaningfully put into broader context of the secretory autophagy-mediated release of (naked virions via) EVs. Sufficient context and references were provided.

A minor suggestion would be to refrain from stating the role of picornavirus security proteins is explored in the study, as only EMCV Leader protein is studied and the data is not discussed in relation to evolutionary conservation of the Leader protein function (or to other security proteins) in picornaviruses.

Authors' response:

We agree with the reviewer that we should refrain from stating that we have examined the role of picornavirus security proteins where we have specifically investigated the function of the EMCV Leader protein. We have modified the abstract of the manuscript to specify that the results obtained in this study pertain only to the EMCV Leader protein. Whether other picornavirus security proteins have similar functions is currently under investigation in our lab.

The study is well-designed, analyzed and presented, and authors use state-of-the-art techniques (e.g. high resolution flow cytometry) to support study conclusions. Still, what is missing in the study is uncovering the mechanism of the observed effect of the Leader protein on secretory autophagy. Is it a direct or indirect effect? Is it connected to other known effects of the Leader dependent on the Zn-finger motif, like well-conserved anti-interferon activity (IFN1 is a known inducer of autophagy), for which mechanism was described?

Authors' response:

Uncovering the mechanism via which the Leader protein promotes secretory autophagy and EV-enclosed virus release is beyond the scope of the current manuscript. We are conducting a follow-up study investigating the interconnection of Leader-induced secretory autophagy and EV-virus release with other known effects of the Leader on the antiviral pathways, and will submit this work in the near future.

Below suggested minor improvements would further strengthen the work:

- Figure 1: Please include definition of TCID50 in the Materials and methods section.

Authors' response:

Throughout the revised manuscript we replaced the term 'TCID50' with 'CCID50', because we consider 'cell culture infectious dose' a better suited term than 'tissue culture infectious dose'. We included the full definition of CCID50 in the Materials & Methods section in lines 557-558.

- Figure 2: The size interval of EV population detected and analyzed by the high resolution flow cytometry is not mentioned but would be important for comparisons to outcomes of other studies (additionally, why were EVs defined as particles 50-300 nm in size, as generally the sizes go up to 1000 nm?).

Authors' response:

We agree with the reviewer that the reported size spectrum of EVs is very broad and ranges beyond the 50-300 nm stated in our original manuscript. On average however, most EVs fall within these more limited dimensions. The high-resolution flow cytometry analysis that we performed cannot be used for sizing of EVs, as light scattering induced by EVs is determined by various factors and not by size alone, as previously reported by our lab¹⁰.

- Figure 3: Was LC3 detected by mass spectrometry? Why were studies of EV-bound LC3 with western blot performed on crude 100K pellets and not EV 1.06-1.1 g/ml density fraction, which represents EVs devoid of naked EMCV virions?

Authors' response:

LC3 was not detected in our EV isolates by mass spectrometry. This is likely due to its small size and relatively low abundance. Because it is not clear in what form LC3 is released during secretory autophagy, we performed western blot analysis for extracellular LC3 on crude ultracentrifugation pellets instead of purified EVs. Our data in Figure 3G suggested that part of the LC3 (likely the lipidated form) was present in EVs because it was protected by proteinase K treatment. In addition, other forms of LC3 may be secreted alongside or associated to the outer surface of EVs. We agree with the reviewer that it is of interest to corroborate these data by assessing the distribution of LC3 in density gradient-separated material. In our revised manuscript, we included western blot analysis for LC3 on density gradient fractions of EMCV-infected samples. The data shows that the lipidated form of LC3 mostly co-segregated with EV-marker proteins, while part of the LC3 was not associated to EVs and remained in high-density bottom fractions of the gradient (new Figure 3H), confirming the proteinase K data. A description of these data can be found in lines 213-215 of the revised manuscript.

- Figure 4: Ratio between autophagosomes vs. autolysosomes is evaluated based on total surface area of EGFP and mCherry positive puncta, therefore I would advise not to interpret this as increase in autophagosomes numbers as increase in total surface area of puncta can be due to increase in number or size of autophagosomes. Additionally, just as a matter of interest, what represents the only green fluorescence observed in some cells; is the mCherry-EGFP-LCS plasmid polycistronic?

Authors' response:

We concur with the reviewer on providing a more accurate interpretation of the confocal microscopy images of the autophagy reporter cells. We have adapted the description of the results

obtained in Figure 4C in lines 237-239 of the revised manuscript, stating that autophagosomes increase either in number or in size upon EMCV-Wt infection.

The mCherry-EGFP-LC3 plasmid is not polycistronic, but codes for a fusion protein with the two fluorescent proteins connected by short peptide linkers. We therefore assume that the compartments that visibly display only EGFP fluorescence do not represent bona fide EGFP+mCherry- compartments, but that they are the result of a difference in detection efficiency between EGFP and mCherry. With the standard laser configuration of the confocal microscope, optimal excitation efficiency could be achieved for EGFP, whereas the excitation efficiency for mCherry is estimated to be only 63% of its optimum. In addition, the quantum yield of mCherry is substantially lower than for EGFP. Given these differences in detection efficiency, we speculate that the mCherry signal is only detected in autophagosomal compartments where there is sufficient accumulation of mCherry-EGFP-LC3, whereas the EGFP signal is more prominently collected for mCherry-EGFP-LC3 in the cytoplasm and in smaller/larger autophagosomes.

- Figure 5: Treatment of EMCV-wt infected cells with rapamycin (stimulator of autophagic flux) re-routed the packaging of viruses in the 10K rather than 100K EVs. What is authors explanation and do these EVs also bind/contain LC3?

Authors' response:

While rapamycin treatment of EMCV-Wt infected cells did not alter the overall amount of EV-enclosed virus, it caused a shift in preferential packaging of the virus in 10K EVs rather than 100K EVs. At this point we can only speculate about explanations for these findings. One possibility is that rapamycin alters the contribution of different EV biogenesis pathways to total EV release, biasing towards the release of larger EVs. Alternatively, the drug may alter the structure or molecular composition of virus-containing EVs in a way that causes them to sediment at lower centrifugal forces.

In response to the reviewer's question about the presence of LC3 in 10K EVs, we performed western blot analysis for LC3 in 10,000xg-pelleted material (Rebuttal figure 1). These data show that LC3 is present in 10K EVs released by EMCV-Wt infected cells, although at a lower level than in 100K EVs. For this reason, we replaced the western blot analyses for the presence of LC3 on 100K EVs (initial submission), with analyses of the total EVs (unseparated 10K + 100K EVs) in Figures 3F-3H and 5B of the revised manuscript.

- Figure 6: Apilimod was shown to induce EV-enclosed virus release in the absence of EMCV Leader activity, but how does it effect the release of FSC^{high} population, representing infectious EV subpopulation?

Authors' response:

We have analyzed the light scattering properties of EVs after apilimod and bafilomycin A treatment of EMCV-L^{Zn} infected cells and included these data as Supplementary figure 6 in the revised manuscript. While apilimod induced the release of LC3 via secretory autophagy and increased the release of infectious virus in EVs (Figure 6), these EVs did not display the FSC^{high} phenotype observed for EVs induced by Leader-competent viruses. This substantiates the idea that the FSC^{high} phenotype corresponds to the Leader-induced EV subset with the highest infectious properties, but is not caused by the presence of viral particles in these EVs. We conclude that treatment with apilimod enables packaging of viruses in EVs, but does not restore all effects of the Leader protein

on the EV phenotype. The description of these data can be found in lines 285-288 of the revised manuscript.

Reviewer #3 (Remarks to the Author):

The work of Van der Grein et al describes a novel role of the EMCV Leader protein in the release of virus enclosed in EVs by a mechanism involving the induction of secretory autophagy. The work is original and relevant to the fields of virology and EVs. The conclusions are supported by the data. The methodology is sound and presented in sufficient detail to be reproducible.

However, I have some comments;

1) Although the experimental settings are explained in enough detail to allow reproducibility, I encourage the authors to submit the data to the EVTRACK website (<http://www.evtrack.org>).

Authors' response:

We thank the reviewer for this advice and have submitted our manuscript to EV-TRACK. The submission of experimental parameters to the EV-TRACK knowledgebase can be accessed via <http://evtrack.org/review.php> using EV-TRACK ID EV220089 and the last name of the first author (Van der Grein).

2) In line 125, the authors state that cells infected with EMCV-L^{zn} mutant do not release 10K EVs. The authors did not show these data. However, I consider it important to show how EV release is changed in cells infected with the mutant virus compared to the control. Characterization of the 10k and 100K EVs in terms of numbers of particles and/or EV markers will be useful to understand how the Leader is changing virus secretion within EVs. Showing differences in infectivity in these fractions and/or their behavior after optiprep separation, although not necessary, would be of added value to having a complete picture of the effects of the Leader on virus secretion inside EVs.

Authors' response:

The reviewer asks for a more complete overview of how infections with wildtype and mutant EMCV change EV release, including effects on distinct 10K and 100K EV subpopulations. In response to this question we added the following data sets:

- We included the new Figure 2A to show the effect of EMCV-Wt and EMCV-L^{zn} infection on the release of 10K and 100K EVs by western blot analyses of several well-known EV marker proteins. These data demonstrate the increase in both 100K and 10K EV release after EMCV-Wt infection, but not following EMCV-L^{zn} infection.
- In conjunction with these data, the new Figure 2B displays the distribution of EV numbers over 10K and 100K subpopulations for both EMCV-Wt and EMCV-L^{zn} infected samples, as assessed by high resolution flow cytometry.
- Figure 1C, which initially only showed infectivity in 100K EV, has now been supplemented with the infectivity in the 10K EV subset for EMCV-Wt and EMCV-L^{zn} infected samples.

Combined, these data substantiate our conclusions that the EV population released by EMCV-L^{zn} infected cells consists almost exclusively of 100K EVs and that no infectious EMCV-L^{zn} can be detected in 10K EVs. In contrast, EMCV-Wt infection induces the release of 10K and 100K EVs and both populations can transfer infections to new cells.

3) Western blots showing 100K EVs in general, or at least the western blot in figure 3A in particular, should show different EV markers according to the guidelines of the International Society for EVs (MISEV2018# 30637094). The Western blots in figures 3G, 5B, and 6A do not contain kDa annotation. The Western blots in figures 3F, 3G and 6A are only representative images of two experiments. Ideally, these westerns should be performed in at least three independent

experiments, and their quantification could be provided. The information about the number of repetitions done for figures 5B and 4A is missing.

Authors' response:

In response to these questions, we made the following adaptations to the manuscript:

- To meet MISEV guidelines, we added analysis of EV marker proteins CD9, CD63, and Flotillin-1 in Figure 2A of our revised manuscript.
- We performed additional repetitions of the western blots depicted in Figures 2A, 3F, 3G, 4A, 5B (N≥3), and Figures 3H and 6A (N=2). For each of these figures, a representative image of multiple independent experiments is presented in the revised manuscript and information about the number of repetitions is included in the relevant figure legends. For data difficult to interpret by eye (changes in cellular LC3 II/I ratios in Figure 4A and Supplementary figure 4C), we also provide the results of quantification.
- We have included the missing kDa annotations in Figures 3G, 5B, and 6A.

4) EV count in Figures 2b and 6b was performed by high-resolution flow cytometry after labeling of the EVs with CFSE. However, this labeling may not label/sufficiently label all EVs subtypes. It would be an added value to include another way of counting EVs, for example using an NTA-based technique.

Authors' response

CFSE labeling:

The reviewer raises a valid and important point. Indeed, generic lipid dyes (e.g. PKH67) are more generally used to stain EVs, and the efficiency with which EVs can be labeled with CFSE varies per cell type. However, an advantage of CFSE is that an enzymatic conversion of the dye within EVs is needed to generate the fluorescent product, thereby reducing interference by non-EV-bound dye (aggregates) often observed with generic lipid dyes. Prior to our current study, we extensively compared CFSE and PKH67 labeling of mock-infected and EMCV-infected HeLa cell EVs. We included an example dataset below (Rebuttal figure 2). For mock-infected HeLa EVs we observed similar staining efficiency. However, we preferred CFSE for studying virus-induced EVs because PKH67 staining efficiency was reduced under infection conditions, whereas CFSE staining efficiency remained stable.

NTA versus flow cytometry:

Previously we published a comparative study on quantification of EVs by high-resolution flow cytometry versus NTA and TRPS¹¹. For highly purified EVs, these techniques yield comparable data, but each technique has pros and cons with regard to sensitivity, specificity and accuracy of EV analysis. For our current study we strongly preferred high-resolution flow cytometry over NTA because the flow cytometric technique requires much lower input EV concentrations. It is important to note that in the short 7 hrs culture period in which we study virus-induced effects, the number of acquired EVs, especially in non-infected control samples, is low. An additional practical reason was that work with picornaviruses needs to be carried out at biosafety level BSL-2, which was possible for the flow cytometric measurements but complicated for NTA.

In an attempt to answer to the request of the reviewer, we found a NanoSight NS3000 NTA instrument that was available at biosafety level BSL-2. To obtain density gradient-purified EVs at the optimal EV concentration range of 10⁷-10⁹ particles/ml for NTA, extensive concentration of EVs from large initial volumes of cell supernatant was necessary. The resulting data on particle concentration and particle sizes in total EV preparations (unseparated 10K + 100K EVs) from mock-infected cells, or cells infected with wildtype or mutant virus, are depicted in Rebuttal figure 3 below. In accordance with high-resolution flow cytometry-based quantification (Figure 2D), the number of EVs released by

cells infected with EMCV-Wt was higher than for mock or EMCV-L^{Zn} infected cells. However, we decided to not include the NTA data in the manuscript because we doubt their reliability. First, the average size and size heterogeneity of mock EVs were higher than expected based on a large number of studies that have reported NTA-based sizing of cell culture-derived EVs. Moreover, the data were highly variable, likely because NTA is not ideally suited to assess heterogeneous samples. We surmised that these virus-induced EV samples are ill-suited for quantification by NTA, due to their low EV concentration as well as the variation and unwanted effects introduced by the elaborate sample preparation. Conversely, high-resolution flow cytometry-based quantification operates within much lower particle concentration ranges and therefore requires less extensive sample processing. We therefore argue that high-resolution flow cytometry is better suited for reliable and robust quantification of EVs in these samples.

5) There is an error in the numbering of Sup Fig 3 in line 264

Authors' response:

We have corrected the numbering of Supplementary Figure 3.

References

1. van der Grein, S. G. *et al.* Picornavirus infection induces temporal release of multiple extracellular vesicle subsets that differ in molecular composition and infectious potential. *PLoS Pathogens* **15**, (2019).
2. Romanova, L. I. *et al.* Antiapoptotic Activity of the Cardiovirus Leader Protein, a Viral “Security” Protein. *Journal of Virology* **83**, 7273–7284 (2009).
3. Balgi, A. D. *et al.* Screen for chemical modulators of autophagy reveals novel therapeutic inhibitors of mTORC1 signaling. *PLoS One* **4**, (2009).
4. Lin, C. *et al.* Amiodarone as an autophagy promotor reduces liver injury and enhances liver regeneration and survival in mice after partial hepatectomy. *Scientific reports* **5**, (2015).
5. Jacquin, E. *et al.* Pharmacological modulators of autophagy activate a parallel noncanonical pathway driving unconventional LC3 lipidation. *Autophagy* **13**, 854-857 (2017).
6. Zhang, Y. *et al.* Autophagy promotes the replication of encephalomyocarditis virus in host cells. *Autophagy* **7**, 613–628 (2011).
7. Wong, J. *et al.* Autophagosome supports coxsackievirus B3 replication in host cells. *Journal of Virology* **82**, 9143-9153 (2008).
8. Abernathy, E. *et al.* Differential and convergent utilization of autophagy components by positive-strand RNA viruses. *PLoS Biology* **17**, (2019).
9. Mauthe, M. *et al.* An siRNA screen for ATG protein depletion reveals the extent of the unconventional functions of the autophagy proteome in virus replication. *Journal of Cell Biology* **214**, 619-635 (2016).
10. van der Vlist, E. J. *et al.* Fluorescent labeling of nano-sized vesicles released by cells and subsequent quantitative and qualitative analysis by high-resolution flow cytometry. *Nature protocols* **7**, 1311–26 (2012).
11. Maas, S. L. N. *et al.* Possibilities and limitations of current technologies for quantification of biological extracellular vesicles and synthetic mimics. *Journal of Controlled Release* **200**, 87-96 (2015).

Rebuttal Figure 1. Presence of LC3 in 10K EVs and 100K EVs.

The presence of LC3 in separate 10K and 100K EV isolates from equal numbers of mock, EMCV-Wt and EMCV-L^{Zn} infected cells was determined by western blotting. 10K EVs were isolated by ultracentrifugation of cell culture supernatants at 10,000xg, and 100K EV were isolated by subsequent ultracentrifugation of 10,000xg supernatants at 100,000xg. Exposure times are indicated in brackets.

Rebuttal Figure 2. EVs released by EMCV-infected cells are efficiently stained with CFSE

Mock-infected or EMCV-Wt infected HeLa cell EVs were stained with CFSE or PKH67 and purified using density gradients for analysis by high resolution flow cytometry. (A) Bar graphs present EV numbers in the 1.04-1.10 g/ml density fractions measured in a fixed time window of 30 seconds. (B) Depicted are representative FSC/CFSE or FSC/PKH67 dot plots of a fixed number of acquired EVs present in the 1.06 g/ml density fraction. Bar graphs display mean fluorescence intensity (MFI) of CFSE or PKH67 staining in the acquired EVs.

Rebuttal Figure 3. Quantification of EV numbers and analysis of EV size by NTA.

Density gradient-purified and concentrated EVs were analyzed by Nanoparticle Tracking Analysis. Indicated are the size distributions, average particle sizes, and particle numbers of EVs released by mock-infected cells, or cells infected with EMCV-Wt or EMCV-L^{Zn}.

Second round of review

Reviewer #1 (Remarks to the Author):

All my questions have been adequately answered. I would like to recommend acceptance.

Reviewer #2 (Remarks to the Author):

Although it would be great to at least hint on the mechanism of the observed effect of the Leader protein on secretory autophagy in the current manuscript, I understand the complexity and workload behind uncovering such mechanisms might prevent that. All my other concerns were adequately addressed by the authors in the resubmitted manuscript. I would suggest the authors add the comment on the "only green fluorescence" to the figure legend 4, as others may be similarly interested in how to interpret that.

Reviewer #3 (Remarks to the Author):

I appreciate that the authors have answered all my comments. I am entirely satisfied with the replies provided by them.

We appreciate the reviewers' recommendation to accept our manuscript. Below, we provide a point-by-point reply to the remaining questions and remarks. Following the guidelines in the author checklist, a new version of the manuscript is supplied with tracked changes, as well as a version with tracked changes accepted.

REVIEWERS' COMMENTS

Reviewer #1 (Remarks to the Author):

All my questions have been adequately answered. I would like to recommend acceptance.

Authors' response: We thank the reviewer for their recommendation.

Reviewer #2 (Remarks to the Author):

Although it would be great to at least hint on the mechanism of the observed effect of the Leader protein on secretory autophagy in the current manuscript, I understand the complexity and workload behind uncovering such mechanisms might prevent that.

All my other concerns were adequately addressed by the authors in the resubmitted manuscript. I would suggest the authors add the comment on the "only green fluorescence" to the figure legend 4, as others may be similarly interested in how to interpret that.

Authors' response: We agree with the reviewer that investigating the mechanisms behind the observed effects of the Leader protein on secretory autophagy and EV-enclosed virus release is of great interest. Indeed, the workload and complexity of the experimentation and analyses required to uncover these mechanisms prevent inclusion in the current manuscript. However, an extensive follow-up study will be submitted in the near future, which examines the interrelation between known effects of the Leader on antiviral pathways, and the orchestration of EV-virus release and secretory autophagy by the Leader.

We have added a statement on our interpretation of the observed EGFP+mCherry- compartments in the legend of Figure 4 of our revised manuscript.

Reviewer #3 (Remarks to the Author):

I appreciate that the authors have answered all my comments. I am entirely satisfied with the replies provided by them.

Authors' response: We thank the reviewer for their response.